# Best-of-N through the Smoothing Lens: KL Divergence and Regret Analysis

**Gholamali Aminian**[¶,*]**, Idan Shenfeld**[†]**, Amir R. Asadi**[‡]**, Ahmad Beirami, Youssef Mroueh**[§]

[¶]The Alan Turing Institute, [†]MIT, [‡]University of Cambridge, [§]IBM

## Abstract

A simple yet effective method for inference-time alignment of generative models is Best-of-$N$ (BoN), where $N$ outcomes are sampled from a reference policy, evaluated using a proxy-reward model, and the highest-scoring one is selected. While prior work argues that BoN is almost optimal in reward vs KL tradeoffs, the effectiveness of BoN depends critically on the quality of the proxy-reward model used for selection. For this purpose, we study BoN through a smooth version known as Soft Best-of-N (SBoN) and develop a theoretical framework to address this gap. We analyze the scaling behavior of BoN by providing bounds on the KL divergence between the SBoN policy and the reference policy, offering insights into how performance varies with the number of samples. We also study the regret gap, i.e., the gap between the expected true-reward under the optimal policy and the SBoN policy. Our theoretical and empirical findings show that smoothing helps SBoN mitigate reward overoptimization, especially when the quality of the proxy-reward is low.

## 1 Introduction

Large language models (LLMs) have transformed machine learning, achieving state-of-the-art results on a variety of tasks. Despite all advancements, LLMs can still generate undesirable outputs, such as toxic or factually incorrect responses. This has made alignment a central goal in modern LLM development (Achiam et al., 2023; Team et al., 2023).

Several post-hoc alignment methods have been proposed to address this challenge, including Reinforcement Learning from Human Feedback (RLHF) (Christiano et al., 2017; Ouyang et al., 2022), SLiC (Zhao et al., 2022), Direct Preference Optimization (Rafailov et al., 2023), controlled decoding (Mudgal et al., 2024) and Best-of-N (BoN) sampling (Stiennon et al., 2020). While these methods differ in their implementation, which range from training-time optimization to test-time selection, they can be viewed as approximating the solution to a KL-regularized reward maximization problem. The optimal solution to this problem is a tilted distribution over responses, which balances reward and proximity to the reference model (Yang et al., 2024).

In BoN as a test-time sampling strategy, given a prompt, $N$ responses are sampled from the reference policy, and the one with the highest proxy-reward sample is selected. Empirically, BoN has been shown to achieve competitive or superior performance in the reward-versus-KL divergence trade-off when compared to RLHF and other alignment methods (Gao et al., 2023a; Mudgal et al., 2024) under *true-reward model*. Furthermore, under certain conditions, it asymptotically approximates the solution to the KL-regularized reward maximization objective (Yang et al., 2024). However, in practice, BoN relies on a learned proxy-reward model, which is an approximation of the true-reward function, to guide this selection. As such, their effectiveness critically depends on both the *proxy-reward model* (estimation error of true-reward) and the *quality of the reference policy*.

Understanding how these two components, the quality of the proxy-reward model and the choice of reference policy, affect the alignment quality of test-time sampling algorithms is essential. There are different measures of alignment quality, including KL divergence [1] between aligned policy and ref-

---

*Corresponding author: gaminian@turing.ac.uk

[1]Unless stated otherwise, all KL divergences are understood to be measured between the aligned policy and the reference policy.

erence policy and the *regret* defined as the gap between the expected true-reward under the optimal (tilted) policy and the alignment policy. Note that minimizing the regret gap is critical to ensuring high-quality outputs and close performance to the optimal policy. Recent work by Gao et al. (2023a) and Hilton et al. (2022) has investigated the scaling laws governing reward model optimization in both reinforcement learning (RL) and BoN settings as a function of KL divergence between aligned policy and reference policy. They empirically demonstrate that, under proxy-reward models, the improvement in expected true-reward, relative to a reference policy, scales proportionally for both RL and BoN policies.

While recent work analyzes BoN under the idealized settings where there is no discrepancy between the proxy-reward and the true-reward (Yang et al., 2024; Beirami et al., 2024; Mroueh, 2024; Huang et al., 2025), our work relaxes this assumption to study the interplay between the reward discrepancy measured through regret and the KL-divergence. We present a theoretical study of **Soft Best-of-$N$ (SBoN)**, a smoothed variant of BoN recently introduced by Mayrink Verdun et al. (2025); Jinnai et al. (2024). Unlike BoN, SBoN draws the final response *probabilistically* from the $N$ candidates, yielding a policy that is tunable with a inverse temperature parameter. Our analysis centres on two metrics:

(a) the Kullback-Leibler divergence between SBoN policy (under the *true* reward or proxy-reward model) and the reference policy, and

(b) the *regret*, i.e. the expected true-reward gap between optimal policy and SBoN policy.

We show how these results specialize to the BoN (as a limit of SBoN for the inverse temperature goes to infinity) and quantify the estimation error incurred by using a *proxy* reward model instead of the true-reward. Finally, we characterize regimes in which SBoN attains *lower* regret bound than BoN when we use the proxy-reward model. Our main contributions are:

- We derive finite-sample upper bounds for KL divergence between the SBoN policy and reference policy, and upper and lower bounds for the regret gap of the SBoN policy. Furthermore, we extend these bounds to BoN. These bounds reveal how the number of responses $N$, proxy-reward model quality, and reference policy model affect performance.

- We quantify cases where SBoN performs better than BoN under an overoptimization scenario where the proxy-reward model is used instead of the true-reward model.

- We provide experimental validation using various proxy-reward models to demonstrate SBoN's advantages in the overoptimization scenario. Furthermore, we provide numerical experiments to evaluate our bounds.

## 2 RELATED WORKS

In this section, we discuss related works on BoN, the theoretical foundation of (Soft) BoN, and overoptimization. More related works for the theoretical foundation of RLHF and smoothing of maximum are provided in the Appendix (App) B.

**Best-of-N:** Despite many recent advancements in alignment, a simple, popular, and well-performing method continues to be the BoN policy (Nakano et al., 2021; Stiennon et al., 2020; Beirami et al., 2024). In fact, Gao et al. (2023b); Mudgal et al. (2024); Eisenstein et al. show that BoN consistently achieves compelling win rate-KL tradeoff curves, often outperforming KL-regularized reinforcement learning and other more complex alignment strategies. LLaMA 2 (Touvron et al., 2023) leverages BoN outputs as teacher signals to further finetune the base model. Mudgal et al. (2024) extend BoN through Q-learning to block-wise BoN decoding. This empirical effectiveness has also inspired research into distilling BoN behavior into standalone models (Amini et al., 2025; Sessa et al., 2024; Qiu et al., 2024). Hughes et al. (2024) utilize BoN as an effective method for jailbreaking, while BoN is also commonly used as a strong baseline for scaling inference-time compute (Brown et al., 2024; Snell et al., 2024). Given the broad success of BoN, we are motivated to theoretically investigate the BoN policies and the effect of the proxy-reward model (reward hacking) and the quality of the reference policy.

**Theoretical Foundation of (Soft) BoN:** KL divergence of BoN is studied in (Beirami et al., 2024; Mroueh, 2024) via information theoretical tools, where the KL divergence of BoN sampling from the reference distribution is bounded by $\log(N) - (N-1)/N$. Scaling laws governing reward

as a function of KL divergence is empirically studied by Gao et al. (2023b) and theoretically formalized by Mroueh (2024). Furthermore, the asymptotic case and the equivalence of BoN to the KL-constrained reinforcement learning solution are studied by Yang et al. (2024) under the assumption of access to optimal reward. Gui et al. (2024) further characterized the win rate–KL gap in the asymptotic regime where a model assigns extremely low likelihoods to successful completions. Furthermore, Sun et al. (2024) accelerated BoN using speculative rejection sampling. The regret of BoN under some assumptions is studied in (Huang et al., 2025). The convergence rate of the SBoN policy to the optimal tilted policy has been analyzed by Mayrink Verdun et al. (2025). Additionally, Geuter et al. (2025) investigate a variant of SBoN that incorporates speculative samples from a small auxiliary model, providing both theoretical and empirical insights. However, the regret gap and KL divergence of SBoN under the overoptimization scenario remain largely unexplored in the existing literature.

## 3 PROBLEM FORMULATION

**Notations:** Upper-case letters denote random variables (e.g., $Z$), lower-case letters denote the realizations of random variables (e.g., $z$), and calligraphic letters denote sets (e.g., $\mathcal{Z}$). All logarithms are in the natural base. The set of probability distributions (measures) over a space $\mathcal{X}$ with finite variance is denoted by $\mathcal{P}(\mathcal{X})$. $\Delta_N$ is $N$-simplex distribution set. The KL divergence between two probability distributions on $\mathbb{R}^d$ with densities $p(x)$ and $q(x)$, such that $q(x) > 0$ when $p(x) > 0$, is $\mathrm{KL}(p\|q) := \int_{\mathbb{R}^d} p(x) \log(p(x)/q(x))\mathrm{d}x$ (with $0.\log 0 := 0$). The total-variation distance is defined as $\mathbb{TV}(p, q) = \frac{1}{2}\int_{\mathcal{X}} |p(x) - q(x)|$. Furthermore, we define chi-square divergence as $\chi^2(p(x)\|q(x)) = \int_{\mathcal{X}} \frac{p^2(x)}{q(x)} - 1$.

**Preliminaries:** Let the finite set[2] of prompts be $\mathcal{X}$ and the discrete finite set of responses be $\mathcal{Y}$. Prompts are drawn from a distribution $\rho$ over $\mathcal{X}$. A (stochastic) policy $\pi \in \Pi$ assigns, for every prompt $x \in \mathcal{X}$, a conditional distribution $\pi(\cdot \mid x)$ over $\mathcal{Y}$; drawing $y \sim \pi(\cdot \mid x)$ yields a response. We treat the supervised-fine-tuned (SFT) model as a *reference policy*, denoted $\pi_{\mathrm{ref}}(\cdot \mid x)$.

### 3.1 REWARD FUNCTION

We consider a (calibrated) true-reward function $r^\star(y, x)$ and a (calibrated) proxy-reward function $\hat{r}(y, x)$, both mapping $\mathcal{Y} \times \mathcal{X} \to [0, 1]$. Throughout the rest of this paper, we assume all reward functions are calibrated (see App. C). For brevity, we refer to them simply as *reward functions*. The true-reward function is estimated via proxy-reward function $\hat{r}(y, x)$ using some preference datasets [3]. As shown by Balashankar et al. (2025), a calibrated reward function satisfies,

- Boundedness: for all $x, y$, we have $\hat{r}(y, x), r^\star(y, x) \in [0, 1]$.
- Uniformity under the reference model: for each prompt $x \in \mathcal{X}$, if $Y \sim \pi_{\mathrm{ref}}(\cdot \mid x)$ then $r(y, x) \sim \mathrm{Unif}(0, 1)$ for $r \in \{\hat{r}, r^\star\}$.

In practice, the proxy-reward model can be fit to a human-labeled preference dataset or to data annotated with true-rewards. Following (Huang et al., 2025), we assume for simplicity that $\hat{r}(y, x)$ is given. We define $\mathcal{Y}^\star_{r^\star}(x) = \arg\max_y r^\star(y, x)$ as the set of maximizers for true-reward. Similarly, we can define $\hat{\mathcal{Y}}_{\hat{r}}(x)$ as the set of maximizers for proxy-reward model. More discussion regarding reward is provided in App. C.

**Assumption 3.1** (Achievable maximum reward). *We assume that for $r \in \{\hat{r}, r^\star\}$, we have $r(\hat{y}(x), x) = 1$ for all $\hat{y}(x) \in \arg\max_y r(y, x)$ and given $x \in \mathcal{X}$.*

In many settings, the reward function attains its maximum at specific responses. In particular, since a large language model (LLM) generates outputs using a finite vocabulary and a bounded number of tokens, the space of possible generations is finite, and thus the assumption holds trivially.

---

[2] For measure-theoretic simplicity and notational convenience, we assume finiteness for the set of prompts. Our results also hold for non-finite set.

[3] In some cases, the reward model is not derived from human preference data. Instead, it is either deterministic (e.g., code execution scores) or provided by an automated classifier (e.g., for toxicity or sentiment).

### 3.2 SBoN Algorithm

Fix a prompt $x \in \mathcal{X}$ and draw $N$ i.i.d. candidates $Y_{1:N} \sim \pi_{\text{ref}}(\cdot \mid x)$. Let $Z \in \{1, \ldots, N\}$ denote the index of the selected response with distribution $P_Z$; write $P_Z(i) = \Pr(Z = i)$. We seek a distribution over indices that maximizes the proxy-reward:

$$\max_{P_Z \in \Delta_N} \mathbb{E}_Z\big[\hat{r}(Y_Z, x)\big].$$

Without regularization, the optimizer is the deterministic Best-of-$N$ (BoN) rule $P_Z = \delta_{i^\star}$ with $i^\star \in \arg\max_i \hat{r}(Y_i, x)$. Because $\hat{r}$ is a *proxy* for the true-reward, this deterministic choice can overoptimize the proxy-reward and get response with lower true-reward. To smooth this, we add an entropy penalty with inverse temperature $\beta > 0$:

$$\max_{P_Z \in \Delta_N} \mathbb{E}_Z\big[\hat{r}(Y_Z, x)\big] + \frac{1}{\beta} H(P_Z).$$

The unique solution is the softmax distribution, $P_Z(i) = \frac{\exp\big(\beta \hat{r}(Y_i, x)\big)}{\sum_{j=1}^N \exp\big(\beta \hat{r}(Y_j, x)\big)}$.

We then sample $Z$ from this distribution and return $Y_Z$. We refer to this sampling rule as **Soft-BoN**, as introduced by Mayrink Verdun et al. (2025). Note that the value of $\beta$ parameter for SBoN is equal to $\beta$ in tilted optimal policy. It is shown by Mayrink Verdun et al. (2025), the SBoN policy can be interpreted as a finite-sample approximation of the tilted optimal policy.

We denote the final policy from SBoN via $\pi_{\hat{r}}^{\text{SBoN}}(y|x)$. Note that for $\beta \to \infty$ and $\beta \to -\infty$, we recover BoN and worst-of-N (WoN) (Balashankar et al., 2025), respectively. Furthermore, for $\beta \to 0$, we recover uniform sampling among the $N$ response samples, which is equivalent to sampling from the reference model $\pi_{\text{ref}}(y|x)$. In (Mayrink Verdun et al., 2025, Lemma 1), the closed form solution of SBoN policy is derived,

$$\pi_{\hat{r}}^{\text{SBoN}}(y|x) = \frac{\pi_{\text{ref}}(y|x) \exp(\beta \hat{r}(y, x))}{Z_{N,\beta}}, \tag{1}$$

where $Z_{N,\beta} = \mathbb{E}\Big[\Big(\frac{1}{N}\big(\exp(\beta \hat{r}(y, x)) + \sum_{i=1}^{N-1} \exp(\beta \hat{r}(Y_i, x)))\big)\Big)^{-1}\Big]^{-1}$.

Similarly, we can define $\pi_{r^\star}^{\text{SBoN}}(y|x)$ based on a true-reward model. For simplicity, we define BoN policies under true-reward and proxy-reward models as $\pi_{r^\star}^{\text{BoN}}(y|x)$ and $\pi_{\hat{r}}^{\text{BoN}}(y|x)$, respectively. In this work, we focus on $\beta \geq 0$. Another motivation for SBoN based on the Gumbel-Max trick is provided in App. E.

### 3.3 Optimal (Tilted) Policy

For a given inverse temperature $\beta > 0$, we seek a policy that remains close to $\pi_{\text{ref}}$ while maximizing expected true-reward, leading to the KL-regularized objective

$$\max_{\pi \in \Pi} \mathbb{E}_{Y \sim \pi(\cdot|x)}\big[r^\star(y, x)\big] - \frac{1}{\beta} \text{KL}\big(\pi(\cdot \mid x) \,\|\, \pi_{\text{ref}}(\cdot \mid x)\big). \tag{2}$$

The unique solution is the optimal *tilted* policy (Korbak et al., 2022b;a; Yang et al., 2024)

$$\pi_{\beta, r^\star}(y|x) = \frac{\pi_{\text{ref}}(y \mid x) \exp\big(\beta r^\star(y, x)\big)}{Z_{r^\star, Y}(x, \beta)}, \tag{3}$$

where $Z_{r^\star, Y}(x, \beta) = \sum_{y \in \mathcal{Y}} \pi_{\text{ref}}(y \mid x) \exp\big(\beta r^\star(y, x)\big)$, is the normalizing (*partition*) function.

Note that, in practice, we do not have access to the closed form of reference policy $\pi_{\text{ref}}(y|x)$ and $r^\star(y, x)$. We can only first estimate the true-reward function via proxy-reward function $\hat{r}(y, x)$ and then sample from $\pi_{\text{ref}}(y|x)$ and compute $\hat{r}(y, x)$ for each individual sample. Finally, we can apply inference-time algorithms, e.g., BoN or SBoN (Mayrink Verdun et al., 2025), where $N$ samples are generated from $\pi_{\text{ref}}(y|x)$ and we choose the sample with the highest proxy-reward function (BoN) or sampled from a distribution (SBoN) using the proxy-reward function. When only a proxy-reward function $\hat{r}(y, x)$ is available, we obtain the analogous partition function $Z_{\hat{r}, Y}(x, \beta)$ and policy $\pi_{\beta, \hat{r}}(\cdot|x)$.

We can also define the optimal policy under the true-reward model as,

$$\pi_{r^\star}^\star(y|x) = \arg\max_\pi \mathbb{E}_{Y \sim \pi(\cdot|x)}[r^\star(Y, x)].$$

Similarly, we can define $\pi_{\hat{r}}^\star(y|x)$ as the optimal policy under the proxy-reward model[4].

As the reward functions (true and proxy) are bounded due to calibration, we can interpret optimal policies as the limit of tilted optimal policies for $r \in \{\hat{r}, r^\star\}$,

$$\pi_{\infty,r}(\cdot|x) := \lim_{\beta \to \infty} \pi_{\beta,r}(\cdot|x)$$

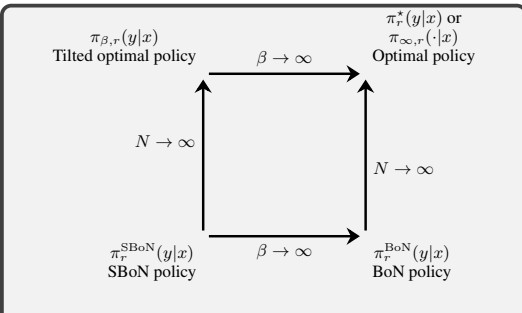

Figure 1: Connections of different policies under reward model $r \in \{\hat{r}, r^\star\}$

where $\pi_{\infty,r^\star}(\cdot|x)$ and $\pi_{\infty,\hat{r}}(\cdot|x)$ place all their probability mass on the maximizers of $r^\star(y, x)$ and $\hat{r}(y, x)$, respectively. The connections between SBoN, BoN, optimal and optimal tilted policies under true or proxy-reward models are shown in Figure 1.

### 3.4 TILTED ERROR

Let's define the tilted error as the tilted average of square estimation error of true-reward function for a given prompt $x$ with parameter $\beta$ as follows,

$$\varepsilon_{\beta,r}(x) := \frac{1}{\beta} \log \left( \mathbb{E}_{Y \sim \pi_{\text{ref}}(y|x)}[e^{\beta(r^\star(Y,x)-\hat{r}(Y,x))^2}] \right). \tag{4}$$

To simplify notation, we adopt the same smoothing parameter $\beta$ for the error definition as is used in the KL and tilted cases. A similar definition of estimation error is introduced in (Yang & Wibisono, 2022). When $\beta = 0$, the definition reduces to the mean-squared error, which is also introduced in (Huang et al., 2025). Letting $\beta \to \infty$ recovers the square of the supremum (infinity) norm ($\|\cdot\|_\infty$) of the estimation error between $r^\star(y, x)$ and $\hat{r}(y, x)$. Therefore, the following properties hold for $\varepsilon_{\beta,r}(x)$,

- The tilted error is bounded, i.e., $\varepsilon_{\beta,r}(x) \in [0, 1]$.
- The tilted average of the estimation error is monotonically increasing in $\beta$.
- $\varepsilon_{\infty,r}(x) := \lim_{\beta \to \infty} \varepsilon_{\beta,r}(x) = \|r^\star(Y, x) - \hat{r}(Y, x)\|_\infty^2$.

We assume that overoptimization regime happens whenever we have $\varepsilon_{\beta,r}(x) > 0$. In more details, $\varepsilon_{\beta,r}(x) > 0$ indicates reward misspecification. As noted in Gao et al. (2023b), misspecification is a necessary precondition for overoptimization, but overoptimization itself refers to the specific regime where the optimization strength is sufficiently high that the degradation due to error outweighs the improvement in the proxy-reward.

We define tilted error using (proxy and true) reward models rather than raw reward models, because our focus is on how rankings change under the proxy. For example, if the proxy-reward is a strictly increasing transform of the true-reward, the ranking is preserved; the Best-of-N (BoN) policy remains optimal and no overoptimization occurs. This behavior cannot be captured when working with the raw (uncalibrated) reward models. Note that in (Huang et al., 2025), raw (uncalibrated) reward models are utilized for error definition.

### 3.5 COVERAGE

For a given reward function $r(x, y)$, we define the tilted policy (softmax policy):

$$\pi_{\beta,r}(y|x) \propto \pi_{\text{ref}}(y|x) \exp(\beta r(y, x)).$$

---

[4]This policy, $\pi_{\hat{r}}^\star(y|x)$ maximizes $\hat{r}_c$, it may be suboptimal or harmful under $r^\star$ due to Goodhart's Law Gao et al. (2023b).

Then, we introduce the coverage of tilted policy with respect to the reference policy as,

$$C_{\beta,r}(x) := \sum_{y \in \mathcal{Y}} \frac{\pi_{\beta,r}^2(y|x)}{\pi_{\text{ref}}(y|x)}. \tag{5}$$

We also define,

$$C_{\infty,\hat{r}}(x) := \lim_{\beta \to \infty} C_{\beta,r}(x).$$

This measure $C_{\beta,r}(x)$ can also be interpreted as a coverage constant, which is standard in KL-regularized policy learning. Furthermore, we can define the coverage of the tilted policy with respect to the reference policy as $\chi^2$-divergence between $\pi_{\beta,r}(y|x)$ and $\pi_{\text{ref}}(y|x)$, i.e., $\chi^2(\pi_{\beta,r}(y|x)\|\pi_{\text{ref}}(y|x))$. It ensures that the reference policy places sufficient probability mass on high-reward responses, thereby guaranteeing that the support of the optimal policy lies within the support of the reference. This prevents cases where optimal outputs are entirely excluded by the reference. Similar notions of coverage have been explored in Huang et al. (2025).

## 3.6 REGRET

For a given policy $\pi(Y|x)$, we define expected true-reward with respect to the policy (a.k.a. value function[5]) as

$$J_{r^\star}(\pi(\,\cdot\,|x)) := \mathbb{E}_{Y \sim \pi(\cdot|x)}[r^\star(Y,x)]. \tag{6}$$

For two policies, $\pi_1(\cdot|x)$ and $\pi_2(\cdot|x)$, we define the gap between these two policies as follows,

$$\Delta_{J_{r^\star}}(\pi_1(\cdot|x), \pi_2(\cdot|x)) := J_{r^\star}(\pi_1(\cdot|x)) - J_{r^\star}(\pi_2(\cdot|x)). \tag{7}$$

We provide an upper bound on the gap of the SBoN solution, which is the gap between $\pi_{r^\star}^\star(\,\cdot\,|x)$ as the optimal policy and $\pi_{\hat{r}}^{\text{SBoN}}(\cdot|x)$,

$$\Delta_{J_{r^\star}}(\pi_{r^\star}^\star(\,\cdot\,|x), \pi_{\hat{r}}^{\text{SBoN}}(\cdot|x)) = J_{r^\star}(\pi_{r^\star}^\star(\,\cdot\,|x)) - J_{r^\star}(\pi_{\hat{r}}^{\text{SBoN}}(\cdot|x)). \tag{8}$$

Regarding regret of the BoN, we consider $\pi_{\hat{r}}^{\text{BoN}}(\cdot|x)$ instead of $\pi_{\hat{r}}^{\text{SBoN}}(\cdot|x)$ in equation 8.

## 4 KL DIVERGENCE ANALYSIS

The KL divergence between the aligned policy and the reference policy, $\text{KL}(\pi_{r^\star}^{\text{BoN}}\|\pi_{\text{ref}})$, is studied by Beirami et al. (2024); Mroueh (2024) from a theoretical perspective. In particular, Beirami et al. (2024) derives an upper bound on KL divergence for BoN policies under the assumptions of a bijective true-reward mapping and a finite output space:

$$\text{KL}(\pi_{r^\star}^{\text{BoN}}(\cdot|x)\|\pi_{\text{ref}}(\cdot|x)) \leq \log(N) - 1 + \frac{1}{N}, \tag{9}$$

Mroueh (2024) relaxes the bijectivity assumption and derives similar bounds using information-theoretic tools. Under some assumptions, the bound in equation 9 is tight. Furthermore, using Pinsker's inequality, in a similar approach to (Mroueh, 2024), we have,

$$\mathbb{E}_{Y \sim \pi_{r^\star}^{\text{SBoN}}(\cdot|x)}[r^\star(Y,x)] \leq 0.5 + \sqrt{\frac{1}{2}\text{KL}(\pi_{r^\star}^{\text{SBoN}}(\cdot|x)\|\pi_{\text{ref}}(\cdot|x))}. \tag{10}$$

Note that equation 10 implies that improvement of expected true-reward relative to the reference policy cannot exceed the square root of the KL divergence. However, the analysis of KL divergence for the SBoN policy under the true-reward model is overlooked. Therefore, we first establish an upper bound on the KL divergence between the SBoN policy under the true-reward model and the reference policy, shedding light on its behavior as a function of the number of samples $N$ and inverse temperature parameter $\beta$. All proof details are deferred to App. G.

---

[5]We can also consider $\mathbb{E}_{X \sim \rho(\cdot)}[J_{r^\star}(\pi(\cdot|X))]$. All of our results also hold for expected version of value function.

**Lemma 4.1.** *The following upper bound holds on KL divergence between SBoN and reference policies for a given prompt $x \in \mathcal{X}$,*

$$\text{KL}(\pi_{r^\star}^{\text{SBoN}}(y|x) \| \pi_{\text{ref}}(y|x)) \leq \log \Big( \frac{N}{1 + (N-1)\exp(-\beta)} \Big). \tag{11}$$

Using Lemma 4.1, we can observe that for BoN, $\beta \to \infty$, we have,

$$\text{KL}(\pi_{r^\star}^{\text{BoN}}(y|x) \| \pi_{\text{ref}}(y|x)) \leq \log(N). \tag{12}$$

Comparing equation 12 with results in (Beirami et al., 2024; Mroueh, 2024), our result is derived from the SBoN asymptotic regime. Note that our bound is looser than the bound on KL divergence in equation 9. In contrast, our bound is general and can be applied to different $\beta$ in SBoN. For $\beta = 0$, where our policy is the reference policy, our bound is tight. It is also important to note that the upper bound in Lemma 4.1 increases with the inverse temperature parameter $\beta$ for fixed $N$.

Recent works by Gao et al. (2023a) and Hilton et al. (2022) empirically demonstrate that, under a true-reward model, the improvement in expected true-reward, relative to a reference policy, scales approximately proportionally to $\sqrt{\text{KL}(\pi_{r^\star}^{\text{BoN}} \| \pi_{\text{ref}})}$ for both RL and BoN policies. It is also observed by Gao et al. (2023b) that models optimized using proxy-rewards can suffer from overoptimization where the learned policy diverges further from the reference, the alignment may degrade. Despite theoretical advances, the KL divergence analysis for SBoN and BoN under the proxy-reward model remains largely unexplored. Therefore, we are interested in investigating the cost we have for estimation error of true-reward via proxy-reward model. For this aim, we first propose the following useful Lemma to study the closeness of the SBoN policy under the true-reward model to the SBoN policy under the proxy-reward model in KL divergence measure.

**Lemma 4.2.** *The following upper bound holds on the KL divergence between the SBoN policies under true-reward and proxy-reward models respectively,*

$$\text{KL}(\pi_{r^\star}^{\text{SBoN}}(\cdot|x) \| \pi_{\hat{r}}^{\text{SBoN}}(\cdot|x)) \leq \frac{N\beta\sqrt{\varepsilon_{\beta,r}(x)}}{1 + (N-1)\exp(-\beta)} \Big( \frac{N\exp(2\beta)}{(N-1)^2} + 1 \Big). \tag{13}$$

Note that for $\beta = 0$, the upper bound in Lemma 4.2 is tight. The result in Lemma 4.2 quantifies the estimation error introduced by substituting a proxy-reward model for the true-reward model.

Next, we compare BoN and SBoN under overoptimization from KL-divergence perspective.

**Remark 4.3** (No overoptimization)**.** *We can observe that for a given $\beta$, if we assume $\varepsilon_{\beta,r}(x) = 0$, we have $\text{KL}(\pi_{r^\star}^{\text{SBoN}}(\cdot|x) \| \pi_{\hat{r}}^{\text{SBoN}}(\cdot|x)) = 0$. Note that, as mentioned in (Gao et al., 2023b), the expected true-reward under the aligned policy, relative to the reference policy, is proportional to the square root of KL divergence. Then a larger KL divergence is desirable in this context, as proposed by (Gao et al., 2023b), the BoN policy is preferred under no overoptimization scenario.*

**Remark 4.4** (Overoptimization)**.** *When $\varepsilon_{\beta,r}(x) > 0$, we have two conflicting goals in both Lemma 4.1 and Lemma 4.2: one suggesting for fixed $N$ that $\beta$ needs to be smaller for better estimation of the true policy by the proxy-reward model one given in Lemma 4.2, and another one suggesting a larger $\beta$ to induce a better KL trade-off based on Gao et al. (2023b). Hence, for a given $N$, there exists an optimal $\beta$ to balance between the estimation error of Lemma 4.2 and the scaling law under the SBoN policy for the true-reward model, Lemma 4.1. In this scenario, SBoN can lead to better tradeoffs than BoN. A similar discussion can be done for fixed $\beta$ and varying $N$. Further analysis using reverse I-projection (Csiszár & Matus, 2003) in large $N$ regime is provided in App. K.*

## 5 REGRET ANALYSIS

In this section, we derive theoretical regret bounds, upper and lower bounds, for SBoN and BoN based on reward models. First, we provide a helpful Lemma regarding the expected coverage assumption that can help us interpret the results of regret for BoN and SBoN. All proof details are deferred to App. H.

**Lemma 5.1.** *Under Assumption 3.1, it holds that $C_{\infty,\hat{r}}(x) = \frac{1}{\sum_i \pi_{\text{ref}}(y_{i,r}^{\max}(x)|x)}$, where $y_{i,r}^{\max}(x) \in \arg\max_y r(y,x)$.*

## 5.1 UPPER BOUND ON REGRET

Now, we derive an upper bound on the regret of SBoN.

**Theorem 5.2** (Upper Bound on Regret of SBoN). *Under Assumption 3.1, the following upper bound holds on the optimal regret gap of the SBoN policy for any $\beta > 0$,*

$$
\Delta_{J_{r^\star}}\left(\pi_{r^\star}^\star(\,\cdot\,|x), \pi_{\hat{r}}^{\mathrm{SBoN}}(\cdot|x)\right) \leq \sqrt{\varepsilon_{\beta,r}(x)}\left(\sqrt{C_{\infty,\hat{r}}(x)} + \sqrt{C_{\infty,r^\star}(x)}\right)
$$
$$
+ 2\sqrt{\frac{1}{2}\log\left(1 + \frac{C_{\infty,\hat{r}}(x) - 1}{N}\right)} + \frac{\log(C_{\infty,r^\star}(x))}{\beta}.
$$

**Regret Bound of BoN Through Smoothing Lens:** We now derive an upper bound on the regret of BoN by taking the asymptotic limit of the regret bound on regret of SBoN in Theorem 5.2.

**Proposition 5.3** (Upper Bound on Regret of BoN). *Under Assumption 3.1, the following upper bound holds on the optimal regret gap of the BoN policy for any $\beta > 0$,*

$$
\Delta_{J_{r^\star}}\left(\pi_{r^\star}^\star(\cdot|x), \pi_{\hat{r}}^{\mathrm{BoN}}(\cdot|x)\right) \leq \sqrt{\varepsilon_{\infty,r}(x)}\left(\sqrt{C_{\infty,\hat{r}}(x)} + \sqrt{C_{\infty,r^\star}(x)}\right)
$$
$$
+ 2\sqrt{\frac{1}{2}\log\left(1 + \frac{C_{\infty,\hat{r}}(x) - 1}{N}\right)}.
$$

**Remark 5.4** (Comparison with (Huang et al., 2025)). *The regret bound for BoN policy grows with the $L_\infty$-norm of the reward-model estimation error. In contrast to the result in (Huang et al., 2025), our bound remains finite whenever the overoptimization error vanishes, i.e., when $\varepsilon_{\infty,\mathrm{r}}(x) = 0$ or $N$ grows. We also derive results based on calibrated reward, instead of raw (uncalibrated) reward models. A full comparison with Huang et al. (2025) is provided in App. J.*

**Overoptimization (asymptotic regime):** Assume that $\varepsilon_{\beta,r}(x) > 0$ for every $\beta > 0$. Letting $N \to \infty$ and invoking Theorem 5.2, we obtain

$$
\Delta_{J_{r^\star}}\left(\pi_{r_\star}^\star(\cdot \mid x), \pi_{\hat{r}}^{\mathrm{SBoN}}(\cdot \mid x)\right) \leq \frac{\log C_{\infty,r^\star}(x)}{\beta}
$$
$$
+ \sqrt{\varepsilon_{\beta,r}(x)}\left(\sqrt{C_{\infty,\hat{r}}(x)} + \sqrt{C_{\infty,r^\star}(x)}\right). \tag{14}
$$

Similarly, for BoN we have,

$$
\Delta_{J_{r^\star}}\left(\pi_{r_\star}^\star(\cdot \mid x), \pi_{\hat{r}}^{\mathrm{BoN}}(\cdot \mid x)\right) \leq \sqrt{\varepsilon_{\infty,r}(x)}\left(\sqrt{C_{\infty,\hat{r}}(x)} + \sqrt{C_{\infty,r^\star}(x)}\right). \tag{15}
$$

**No overoptimization:** Assume that the overoptimization vanishes, i.e. $\varepsilon_{\beta,r}(x) = 0$ for every $\beta \in [0, \infty)$. Then the optimality gaps of the SBoN and BoN policies satisfy

$$
\Delta_{J_{r^\star}}\left(\pi_{r_\star}^\star(\cdot \mid x), \pi_{\hat{r}}^{\mathrm{SBoN}}(\cdot \mid x)\right) \leq 2\sqrt{\frac{1}{2}\log\left(1 + \frac{C_{\infty,\hat{r}}(x) - 1}{N}\right)} + \frac{\log C_{\infty,r^\star}(x)}{\beta}, \tag{16}
$$

$$
\Delta_{J_{r^\star}}\left(\pi_{r_\star}^\star(\cdot \mid x), \pi_{\hat{r}}^{\mathrm{BoN}}(\cdot \mid x)\right) \leq 2\sqrt{\frac{1}{2}\log\left(1 + \frac{C_{\infty,\hat{r}}(x) - 1}{N}\right)}. \tag{17}
$$

## 5.2 LOWER BOUND ON REGRET

In this section, we complement the regret upper bounds (Theorem 5.2 and Proposition 5.3) with lower bounds that hold for any finite $N$ and fixed $\beta \geq 0$. For the lower bound, the following assumptions are needed.

**Assumption 5.5** (Margin Assumption). *Let $\gamma(x) = 1 - \sup_{y \notin \mathcal{Y}_{r^\star}^\star(x)} r^\star(y, x)$. We assume that $\gamma(x) \in (0, 1)$.*

Note that Assumption 5.5 for strictly positive lower bound is needed.

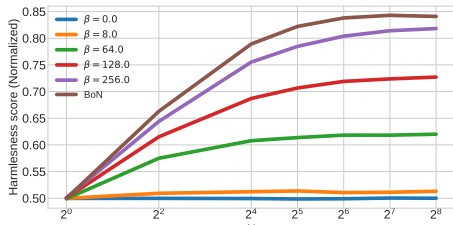 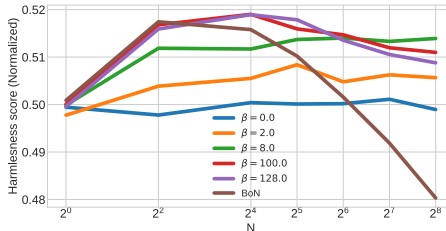

Figure 2: Soft Best-of-N experiment using a strong reward model (Left) and a weak one (Right). Number of samples versus Harmlessness score (higher is better). When the performance of the reward model is poor, BoN can lead to overoptimization, while the SBoN can help to mitigate it.

**Theorem 5.6** (Lower Bound on Regret of SBoN). *Under the same Assumptions in Theorem 5.2 and Lemma H.3, the following lower bound holds on the regret of SBoN policy under the proxy-reward model,*

$$\Delta_{J_{r^\star}}\left(\pi_{r^\star}^\star(\cdot|x), \pi_{\hat{r}}^{\mathrm{SBoN}}(\cdot|x)\right)$$

$$\geq \gamma(x)\left((1 - C_{\infty,r^\star}(x)^{-1})^N + \frac{N((1 - C_{\infty,r^\star}(x)^{-1}) - (1 - C_{\infty,r^\star}(x)^{-1})^N)}{N + (\exp(\beta) - 1)(N - 1)}\right) \qquad (18)$$

$$- \min\left(\sqrt{1 - \exp\left(-U(N,\beta)\right)}, \frac{N}{1 + (N-1)\exp(-\beta)}\sqrt{\varepsilon_{\beta,r}(x)}\right)$$

*where $U(N,\beta) = \frac{N\beta\sqrt{\varepsilon_{\beta,r}(x)}}{1+(N-1)\exp(-\beta)}\left(\frac{N\exp(2\beta)}{(N-1)^2} + 1\right)$.*

Using similar approach to Proposition 5.3, we can derive the lower bound on regret of the BoN policy through the lens of smoothing.

**Proposition 5.7** (Lower Bound on Regret of BoN). *Under the same assumptions in Theorem 5.6, the following lower bound holds on regret of BoN policy,*

$$\Delta_{J_{r^\star}}\left(\pi_{r^\star}^\star(\cdot|x), \pi_{\hat{r}}^{\mathrm{BoN}}(\cdot|x)\right) \geq \gamma(x)(1 - C_{\infty,r^\star}(x)^{-1})^N - \min(1, N\sqrt{\varepsilon_{\infty,r}(x)}). \qquad (19)$$

Under no overoptimization ($\varepsilon_{\infty,r}(x) = 0$), due to $(1 - C_{\infty,r^\star}(x)^{-1})^N \geq \exp(\frac{-N}{C_{\infty,r^\star}(x)-1})$, the lower bound on regret of BoN is $O(\exp(-N))$. Furthermore, in App. I.2, we present a numerical example to illustrate the positivity of the lower bounds under certain conditions.

## 5.3 DISCUSSION

This section analyzes the performance of BoN and SBoN in the presence and absence of overoptimization.

**Remark 5.8** (Overoptimization). *Considering the upper bounds in equation 14 and equation 15, we define,*

$$g(\beta) = \beta\left(\sqrt{\varepsilon_{\infty,r}(x)} - \sqrt{\varepsilon_{\beta,r}(x)}\right), \qquad \beta \geq 0.$$

*Because $g(0) = g(\infty) = 0$ and $g(\beta) \geq 0$ for all $\beta$, there exists at least one maximizer $\beta^\star \in (0, \infty)$ such that $g(\beta^\star) = \max_{\beta \geq 0} g(\beta)$.*

*If $\frac{\log C_{\infty,r^\star}(x)}{\sqrt{C_{\infty,\hat{r}}(x)} + \sqrt{C_{\infty,r^\star}(x)}} \leq g(\beta^\star)$, then the upper bound in equation 14 does not exceed equation 15, and hence the bound on the regret of the SBoN policy is* tighter *than the bound on the regret of the BoN policy under the proxy-reward model. An analogous comparison can be carried out for any fixed $\beta$ and changing $N$. A numerical example for the existence of optimal $\beta$ based on this approach is provided in App. I.2.*

**Remark 5.9** (No overoptimization). *By Lemma 5.1, $C_{\infty,r^\star}(x) \geq 1$; consequently, the bound in equation 17 is tighter than the bound in equation 16. Similar discussion can be provided for lower bound.*

**Remark 5.10** (Quality of reference policy). *The upper bounds in Proposition 5.3 (or Theorem 5.2) depend on two quantities,*

$$C_{\infty,r^\star}(x) = \frac{1}{\sum_i \pi_{\mathrm{ref}}\big(y_{i,r^\star}^{\max}(x) \mid x\big)}, \quad and \quad C_{\infty,\hat{r}}(x) = \frac{1}{\sum_i \pi_{\mathrm{ref}}\big(y_{i,\hat{r}}^{\max}(x) \mid x\big)},$$

*which represent its quality under the true-reward model and proxy-reward model, respectively. A larger value for these quantities implies that the reference model rarely generates optimal responses, thereby degrading performance. We can observe that upper bounds increase by increasing $C_{\infty,r^\star}(x)$ and $C_{\infty,\hat{r}}(x)$.*

## 6 EXPERIMENTS

To support our theoretical analysis, we conducted experiments comparing Soft Best-of-N (SBoN) across different regularization strengths and reward model qualities. We used the Olmo-2 1B model (OLMo et al., 2024) as the generator and prompts from the Attaq dataset (Kour et al., 2023). For each prompt, we generated multiple responses and selected one using SBoN with varying inverse temperature values $\beta$. We ran two experimental conditions: one using a strong proxy-reward model (ArmoRM 8B (Wang et al., 2024)) which is close to true-reward model, and another using a weaker proxy-reward model (Beaver 7B RM (Dai et al., 2023)). We use LLM-as-a-Judge Zheng et al. (2023) as our $r^\star$. To match our theoretical setting, we perform empirical calibration of each reward model by sampling 256 responses for every query and calculate the quantiles. As shown in Figure 2, when the reward model is weak, performance degrades for large $N$ due to reward hacking. However, the smoothing in SBoN helps mitigate this degradation. This observation is also aligned with our theoretical analysis and discussion in Section 4, where under overoptimization there exists a $\beta$ for a given $N$ which outperforms BoN. For more details, see App. I. We also studied the behavior of our upper bound on the KL divergence between the SBoN policy and the reference policy, Lemma 4.1, in App. I.2. More experiments with a medium weak reward model are provided in App. I.1.

## 7 CONCLUSION AND FUTURE WORK

In this work, we establish a theoretical foundation for alignment strategies based on Soft Best-of-N (SBoN) and Best-of-N (BoN) policies. Specifically, we derive upper bounds on the KL divergence between the aligned policy such as SBoN or BoN and the reference policy. We also studied the regret gap between the optimal policy and the aligned policy, e.g., BoN and SBoN policies. We further analyze how errors in reward estimation affect performance in both KL divergence and regret gap. Notably, both our theoretical analysis and empirical results demonstrate that, under a proxy-reward model where overoptimization happens, SBoN performs better than BoN under some conditions.

Existing literature Beirami et al. (2024); Mroueh (2024), suggests a tighter upper bound on the KL divergence for BoN policy. Our derived result, however, provides a looser bound for BoN. Consequently, an interesting direction for future research is the derivation of an upper bound that is asymptotically tight for the BoN policy. Furthermore, while our current analysis utilizes a shared $\beta$ for both the tilted error and the SBoN policy, exploring decoupled inverse temperature parameters for these components remains a promising avenue for future study.

## ACKNOWLEDGMENT

Gholamali Aminian acknowledges the support of the UKRI Prosperity Partnership Scheme (FAIR) under EPSRC Grant EP/V056883/1 and the Alan Turing Institute. Amir R. Asadi is supported by Leverhulme Trust grant ECF-2023-189 and Isaac Newton Trust grant 23.08(b).

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

# Appendix

## Table of Contents

# A    TABLE OF NOTATIONS

All notations are summarized in Table 1.

| Notation | Definition |
|---|---|
| $r^\star(y, x)$ | (Calibrated) true-reward model |
| $\hat{r}(y, x)$ | (Calibrated) proxy-reward model |
| $N$ | Number of responses |
| $x$ | Prompt |
| $\beta$ | Temperature (regularization coefficient) |
| $\pi_{\beta,r}(y\|x)$ | Tilted policy under reward function $r$ |
| $\pi_r^{\mathrm{SBoN}}(y\|x)$ | SBoN policy under reward function $r$ |
| $\pi_r^{\mathrm{BoN}}(y\|x)$ | BoN policy under reward function $r$ |
| $\pi_{\mathrm{ref}}$ | Reference policy |
| $\mathrm{KL}(p\|q)$ | KL divergence between $p$ and $q$ distributions |
| $\pi_r^\star(y\|x)$ | Optimal policy |
| $C_{\beta,r}(x)$ | Coverage constant under reward model $r$ |
| $\mathcal{Y}_{r^\star}^\star(x)$ | Set of maximizers for $r^\star(y, x)$ |
| $\varepsilon_{\beta,r}(x)$ | Tilted Error |

Table 1: Summary of notations in the paper

# B    OTHER RELATED WORKS

**Smoothing of Maximum:** Approximating the maximum operator using a smoothed or softmax-based surrogate is a widely adopted technique in machine learning. This approach is particularly useful in settings where the hard maximum is non-differentiable or leads to unstable optimization. For instance, in robust regression, smooth approximations to the max operator are used in min-max formulations to achieve tractable optimization under distributional shifts (Wang et al., 2013; Li et al., 2023). In sequential decision-making, similar ideas appear in risk-sensitive control, Q-learning and off-policy learning, where the softmax of Q-values leads to stochastic policies that balance exploration and exploitation (Howard & Matheson, 1972; Borkar, 2002; Behnamnia et al., 2025). In convex and non-convex optimization, smoothing the maximum objective has been shown to improve convergence properties (Kort & Bertsekas, 1972; Pee & Royset, 2011; Liu & Theodorou, 2019). The Soft Best-of-N (SBoN) framework, (Mayrink Verdun et al., 2025; Khanov et al., 2024; Jinnai et al., 2024), leverages this principle by replacing the hard selection of the highest-reward sample with a softmax-weighted sampling distribution. Regarding the SBoN, the empirical version of SBoN is introduced by (Khanov et al., 2024) as ARGS-stochastic, where a token from a probability distribution among the top-k candidate tokens is chosen. Then, the regularized version of BoN, which can be represented as SBoN, is discussed by (Jinnai et al., 2024). Given the broad success of SBoN, we are motivated to theoretically investigate the SBoN policies and the effect of the proxy-reward model (reward hacking) and the quality of the reference policy.

**Theoretical Foundation of RLHF:** Several works have studied the theoretical underpinnings of reverse KL-regularized RLHF, particularly in terms of sample complexity (Zhao et al., 2024; Xiong et al., 2024; Song et al., 2024; Zhan et al., 2023; Ye et al., 2024; Aminian et al., 2025). Note that, as the sampling distributions in BoN and SBoN are different, we cannot apply RLHF analysis to these sampling strategies. Therefore, it is needed to develop new foundations for BoN and SBoN.

**Overoptimization.** Alignment methods are widely known to suffer from overoptimization, also known as misspecification, reward hacking, or Goodhart Law, where optimizing against a proxy-reward model leads to degraded performance compared to the true-reward model (Amodei et al., 2016; Casper et al., 2023; Gao et al., 2023b). This issue is particularly pronounced in inference-time alignment methods such as BoN, where an increasing number of responses $N$ makes the overoptimization problem worse (Huang et al., 2025; Stroebl et al., 2024; Gao et al., 2023b). Huang et al. (2025) theoretically demonstrate that the BoN policy suffers from overoptimization when $N$ is large, given a fixed estimation error in the reward model, and propose a solution based on a $\chi^2$-regularized

framework. Other approaches to mitigating this issue include ensembling strategies (Coste et al.; Eisenstein et al.) and regularization techniques (Ichihara et al.). In a concurrent line of work, Khalaf et al. (2025) introduce the Best-of-Poisson method to reduce overoptimization in inference-time algorithms. The overoptimization in BoN and SBoN is also studied by Khalaf et al. (2025) and a principled hedging framework is proposed to mitigate the overoptimization. In contrast, we study overoptimization in inference-time alignment methods SBoN and BoN from the perspectives of regret gap and KL divergence analysis.

## C CALIBRATED REWARD

Inspired by (Balashankar et al., 2025), in this section, we provide more details regarding calibrated reward. A standard metric for evaluation of models is the *win-rate* relative to a base policy $\pi_{\mathrm{ref}}$ (Stiennon et al., 2020; Gao et al., 2023b). For a prompt $x$ and responses $y, z$, define the win random variable under raw (uncalibrated) reward $r_{\mathrm{uc}}$ as

$$w_r(y, z \mid x) = \mathbf{1}\{r_{\mathrm{uc}}(y, x) > r_{\mathrm{uc}}(z, x)\} + \tfrac{1}{2}\mathbf{1}\{r_{\mathrm{uc}}(y, x) = r_{\mathrm{uc}}(z, x)\}.$$

**Definition C.1** (Calibrated reward)**.** *The calibrated reward of $y$ under policy $\pi$ is its expected win-rate probability against $z \sim \pi(\cdot \mid x)$:*

$$r_{\mathrm{c},\pi}(y, x) \coloneqq \mathbb{E}_{z\sim\pi(\cdot\mid x)}[w_r(y, z \mid x)].$$

In practice, we consider $\pi = \pi_{\mathrm{ref}}$, therefore we denote calibrated reward via $r(y, x)$ under reference policy. In the following, we provide some reasons for choosing calibrated reward instead of raw (uncalibrated) reward in our work,

- **Matches win-rate evaluation.** For any policies $\pi_1, \pi_2$,

$$\mathrm{W}_r(\pi_1 \succ \pi_2 \mid x) \coloneqq \mathbb{E}_{y\sim\pi_1(\cdot\mid x)}[r_{c,\pi_2}(y, x)],$$

  where $\mathrm{W}_r(\pi_1 \succ \pi_2 \mid x)$ is standard win-rate. So maximizing $\mathbb{E}_{Y\sim\pi}[r_{c,\pi_{\mathrm{ref}}}(y, x)]$ directly optimizes standard win rate vs. the base model.
- **Invariance to score scaling.** If $m$ is strictly increasing and $r' = m \circ r$, then

$$r'(y, x) \;=\; r(y, x),$$

  making the target robust to arbitrary monotone reparameterizations of the reward (e.g., affine rescaling, temperature).
- **Unified, probabilistic scale.** For $y \sim \pi_{\mathrm{ref}}(\cdot \mid x)$,

$$r(y, x) \sim \mathrm{Unif}[0, 1],$$

  independent of both $r$ and $\pi_{\mathrm{ref}}$. This normalizes per-prompt reward scales and interprets scores as win probabilities.

## D SUMMARY OF KL DIVERGENCE RESULTS

In Table 2 , we summarize results on KL divergences between the aligned and reference policies, along with corresponding upper bounds for both SBoN and BoN policies. Furthermore, in Table 3, we summarize results on KL divergences between aligned policies under true and proxy-reward models, along with upper bounds for SBoN and tilted policies.

## E GUMBEL-MAX TRICK

We also provide an interpretation for SBoN from the Gumbel-Max trick. An alternative way to sample $Z$ from
$$\Pr(Z = i) \;\propto\; \exp\!\big(\beta\,\hat{r}(Y_i, x)\big)$$
is via the Gumbel-Max trick. We can draw independent Gumbel-distributed random variables $G_i \sim \mathrm{Gumbel}(0, 1), i = 1, \ldots, n$, and then set

$$Z = \arg\max_{i\in\{1,\ldots,N\}}\left[\hat{r}(Y_i, x) + \frac{G_i}{\beta}\right].$$

Table 2: KL divergences between the aligned and reference policies, along with corresponding upper bounds for both SBoN and BoN policies.

| KL divergence Term | Theorem / Lemma | Upper Bound |
|---|---|---|
| $\mathrm{KL}\big(\pi_{r^\star}^{\mathrm{SBoN}}(\cdot|x) \,\|\, \pi_{\mathrm{ref}}(\cdot|x)\big)$ | Lemma 4.1 | $\log\left(\frac{N}{1+(N-1)\exp(-\beta)}\right)$ |
| $\mathrm{KL}(\pi_{r^\star}^{\mathrm{BoN}}(\cdot|x)\|\pi_{\mathrm{ref}}(\cdot|x))$ | Theorem 3.1 in (Beirami et al., 2024) and Theorem 1 in (Mroueh, 2024) | $\log(N) - 1 + 1/N$ |

Table 3: KL divergences between aligned policies under true and proxy-reward models, along with upper bounds for SBoN and tilted policies.

| KL divergence Term | Theorem / Lemma | Upper Bound |
|---|---|---|
| $\mathrm{KL}\big(\pi_{r^\star}^{\mathrm{SBoN}}(\cdot|x) \,\|\, \pi_{\hat{r}}^{\mathrm{SBoN}}(\cdot|x)\big)$ | Lemma 4.2 | $\frac{N\beta\sqrt{\varepsilon_{\beta,r}(x)}}{1+(N-1)\exp(-\beta)}\left(\frac{N\exp(2\beta)}{(N-1)^2}+1\right)$ |
| $\mathrm{KL}\big(\pi_{\beta,r^\star}(\cdot|x) \,\|\, \pi_{\beta,\hat{r}}(\cdot|x)\big)$ | Lemma F.8 | $2\beta\sqrt{\varepsilon_{\beta,r}(x)}\left(\sqrt{\frac{\mathbb{E}[\exp(2\beta\hat{r}(Y,x)]}{\mathbb{E}^2[\exp(\beta\hat{r}(Y,x))]}}+\sqrt{\frac{\mathbb{E}[\exp(2\beta r^\star(Y,x)]}{\mathbb{E}^2[\exp(\beta r^\star(Y,x))]}}\right)$ |

By properties of the Gumbel distribution, this yields exactly the same softmax sampling law, without needing to compute the normalizing factor $\sum_{j=1}^{N}\exp(\beta\hat{r}(Y_j,x))$ explicitly (Gumbel, 1954). When $\beta \to \infty$, the effect of the Gumbel noises vanishes and the sampling strategy reduces to BoN.

## F  TECHNICAL TOOLS

We denote the set maximizers of the proxy-reward function via $\hat{\mathcal{Y}}(x) = \{\hat{y}_j(x)\}_{j=1}^{m(x)}$.

We introduce the functional derivative, see Cardaliaguet et al. (2019).

**Definition F.1.** *(Cardaliaguet et al., 2019)  A functional $U : \mathcal{P}(\mathbb{R}^n) \to \mathbb{R}$ admits a functional derivative if there is a map $\frac{\delta U}{\delta m} : \mathcal{P}(\mathbb{R}^n) \times \mathbb{R}^n \to \mathbb{R}$ which is continuous on $\mathcal{P}(\mathbb{R}^n)$ and, for all $m, m' \in \mathcal{P}(\mathbb{R}^n)$, it holds that*

$$U(m') - U(m) = \int_0^1 \int_{\mathbb{R}^n} \frac{\delta U}{\delta m}(m_\beta, a)\,(m'-m)(da)\,\mathrm{d}\beta,$$

*where $m_\beta = m + \beta(m' - m)$.*

**Definition F.2** (Sensitivity of a policy). *We also define the sensitivity of a policy $\pi_r(y|x)$, which is a function of reward function $r(y,x)$, with respect to the reward function as*

$$\frac{\partial\pi}{\partial r}(r) := \lim_{\Delta r \to 0} \frac{\pi_r(y|x) - \pi_{r+\Delta r}(y|x)}{\Delta r}. \tag{20}$$

**Lemma F.3** (Kantorovich-Rubenstein duality of total variation distance, see (Polyanskiy & Wu, 2022)). *The Kantorovich-Rubenstein duality (variational representation) of the total variation distance is as follows:*

$$\mathbb{TV}(m_1, m_2) = \frac{1}{L}\sup_{g \in \mathcal{G}_L}\{\mathbb{E}_{Z\sim m_1}[g(Z)] - \mathbb{E}_{Z\sim m_2}[g(Z)]\}, \tag{21}$$

*where $\mathcal{G}_L = \{g : \mathcal{Z} \to \mathbb{R}, \|g\|_\infty \leq L\}$.*

**Lemma F.4** (Lemma D.6 in (Aminian et al., 2025)). *Consider the softmax policy, $\pi_r^\beta(y|x) \propto \pi_{\mathrm{ref}}(y|x)\exp(\beta r(y,x))$. Then, the sensitivity of the policy with respect to the reward function is*

$$\frac{\partial\pi_r^\beta}{\partial r}(r) = \beta\pi_r^\beta(y|x)(1 - \pi_r^\beta(y|x)).$$

**Lemma F.5** (Pinskers Inequality (Canonne, 2022)). *For $m_1$ and $m_2$, we have,*

$$\mathbb{TV}(m_1, m_2) \leq \sqrt{\frac{1}{2}\mathrm{KL}(m_2\|m_1)}. \tag{22}$$

The following Lemmata are useful for our technical proofs.

**Lemma F.6.** *The following upper bound holds,*

$$\log\Big(\frac{Z_{r^\star,Y}(x,\beta)}{Z_{\hat{r},Y}(x,\beta)}\Big) \leq \beta\sqrt{\varepsilon_{\beta,r}(x)}\sqrt{C_{\beta,r^\star}(x)}. \tag{23}$$

*Proof.*

$$\begin{aligned}
\frac{Z_{\hat{r},Y}(x,\beta)}{Z_{r^\star,Y}(x,\beta)} &= \frac{\sum_{\mathcal{Y}}\exp(\beta\hat{r}(y,x))\pi_{\text{ref}}(y|x)}{\sum_{\mathcal{Y}}\exp(\beta r^\star(y,x))\pi_{\text{ref}}(y|x)} \\
&= \frac{\sum_{\mathcal{Y}}\exp(\beta(\hat{r}(y,x) - r^\star(y,x)))\exp(\beta r^\star(y,x))\pi_{\text{ref}}(y|x)}{\sum_{\mathcal{Y}}\exp(\beta r^\star(y,x))\pi_{\text{ref}}(y|x)} \\
&= \sum_{\mathcal{Y}}\pi_{\beta,r^\star}(y|x)\exp(\beta(\hat{r}(y,x) - r^\star(y,x)))
\end{aligned} \tag{24}$$

Due to convexity of $-\log(\cdot)$ and using Cauchy-Schwarz inequality, we have,

$$\begin{aligned}
-\log\Big(\frac{Z_{\hat{r},Y}(x,\beta)}{Z_{r^\star,Y}(x,\beta)}\Big) &\leq \beta\sum_{\mathcal{Y}}\pi_{\beta,r^\star}(y|x)(r^\star(y,x) - \hat{r}(y,x)) \\
&\leq \beta\sum_{\mathcal{Y}}\frac{\pi_{\beta,r^\star}(y|x)}{\pi_{\text{ref}}(y|x)}(r^\star(y,x) - \hat{r}(y,x))\pi_{\text{ref}}(y|x) \\
&\leq \beta\sqrt{\sum_{\mathcal{Y}}(r^\star(y,x) - \hat{r}(y,x))^2\pi_{\text{ref}}(y|x)}\sqrt{\sum_{\mathcal{Y}}\frac{\pi_{\beta,r^\star}^2(y|x)}{\pi_{\text{ref}}(y|x)}} \\
&= \sqrt{\beta}\sqrt{\sum_{\mathcal{Y}}\log\big(\exp\big(\beta(r^\star(y,x) - \hat{r}(y,x))^2\big)\big)\pi_{\text{ref}}(y|x)}\sqrt{C_{\beta,r^\star}(x)} \\
&\leq \beta\sqrt{\frac{1}{\beta}\log\big(\sum_{\mathcal{Y}}\exp\big(\beta(r^\star(y,x) - \hat{r}(y,x))^2\big)\pi_{\text{ref}}(y|x)\big)}\sqrt{C_{\beta,r^\star}(x)} \\
&= \beta\sqrt{\varepsilon_{\beta,r}(x)}\sqrt{C_{\beta,r^\star}(x)},
\end{aligned} \tag{25}$$

$\square$

**Lemma F.7.** *The following holds,*

$$\text{KL}(\pi_{r^\star}^\star(\,\cdot\,|x)\|\pi_{\text{ref}}(\cdot|x)) \leq \log(C_{\infty,r^\star}(x)). \tag{26}$$

*Proof.* Note that, we have,

$$\begin{aligned}
\text{KL}(\pi_{r^\star}^\star(\,\cdot\,|x)\|\pi_{\text{ref}}(\cdot|x)) &\leq \log\Big(\mathbb{E}_{Y\sim\pi_{r^\star}^\star(\,\cdot\,|x)}\Big[\frac{\pi_{r^\star}^\star(\,\cdot\,|x)}{\pi_{\text{ref}}(\cdot|x)}\Big]\Big) \\
&\leq \log(C_{\infty,r^\star}(x)).
\end{aligned} \tag{27}$$

$\square$

**Lemma F.8.** *The following upper bound holds,*

$$\text{KL}(\pi_{\beta,r^\star}(y|x)\|\pi_{\beta,\hat{r}}(y|x)) \leq \beta\sqrt{\varepsilon_{\beta,r}(x)}\big(\sqrt{C_{\beta,r^\star}(x)} + \sqrt{C_{\beta,\hat{r}}(x)}\big). \tag{28}$$

*Proof.*

$$\begin{aligned}
\text{KL}(\pi_{\beta,r^\star}(y|x)\|\pi_{\beta,\hat{r}}(y|x)) &= \sum_{\mathcal{Y}}\pi_{\beta,r^\star}(y|x)\log\Big(\frac{\pi_{\beta,r^\star}(y|x)}{\pi_{\beta,\hat{r}}(y|x)}\Big) \\
&= \beta\sum_{\mathcal{Y}}(r^\star(y,x) - \hat{r}(y,x))\pi_{\beta,r^\star}(y|x) + \log(Z_{\hat{r},Y}(x,\beta)/Z_{r^\star,Y}(x,\beta)) \\
&\leq \beta\sqrt{\varepsilon_{\beta,r}(x)}\big(\sqrt{C_{\beta,r^\star}(x)} + \sqrt{C_{\beta,\hat{r}}(x)}\big),
\end{aligned}$$

$$\tag{29}$$

where the final inequality holds due to Lemma F.6 and applying Cauchy-Schwarz inequality. $\qquad\square$

**Lemma F.9.** *Suppose that $f(Z) \in [0, Z_{\max}]$, $\mathcal{Z}_{\max} = \{z_{m,i}\}_{i=1}^m$ is the set of maximizers of $f(Z)$, i.e., $f(z) = Z_{\max}$ for $z \in \mathcal{Z}_{\max}$. Then we have,*

$$\lim_{\beta \to \infty} \frac{\mathbb{E}[\exp(2\beta f(Z))]}{\mathbb{E}[\exp(\beta f(Z))]^2} = \frac{1}{\sum_{z \in \mathcal{Z}_{\max}} P(Z = z)}. \tag{30}$$

*Proof.*

$$\frac{\mathbb{E}[\exp(2\beta f(Z))]}{\mathbb{E}[\exp(\beta f(Z))]^2} = \frac{\mathbb{E}[\exp(2\beta(f(Z) - Z_{\max}))]}{\mathbb{E}[\exp(\beta(f(Z) - Z_{\max}))]^2} \tag{31}$$

$$\frac{\sum_j P(Z = z_j) \exp(2\beta(f(z_j) - Z_{\max}))}{(\sum_j P(Z = z_j) \exp(\beta(f(z_j) - Z_{\max})))^2} \tag{32}$$

Now, we have,

$$\lim_{\beta \to \infty} \frac{\mathbb{E}[\exp(2\beta f(Z))]}{\mathbb{E}[\exp(\beta f(Z))]^2} \tag{33}$$

$$= \lim_{\beta \to \infty} \frac{\sum_j P(Z = z_j) \exp(2\beta(f(z_j) - Z_{\max}))}{(\sum_j P(Z = z_j) \exp(\beta(f(z_j) - Z_{\max})))^2} \tag{34}$$

$$= \frac{\sum_{z \in \mathcal{Z}_{\max}} P(Z = z)}{(\sum_{z \in \mathcal{Z}_{\max}} P(Z = z))^2} \tag{35}$$

$$= \frac{1}{\sum_{z \in \mathcal{Z}_{\max}} P(Z = z)}, \tag{36}$$

where we used the fact that $\lim_{\beta \to \infty} \exp(\beta(z_j - Z_{\max})) = 0$ for $z_j < Z_{\max}$. $\qquad\square$

**Lemma F.10** (Theorem 1 in (Mayrink Verdun et al., 2025)). *For $\beta > 0$, and $N \geq 1$, we have,*

$$\mathrm{KL}(\pi_{\beta,r^\star}(\cdot|x) \| \pi_{r^\star}^{\mathrm{SBoN}}(y|x)) \leq \log\left(1 + \frac{C_{\beta,r^\star}(x) - 1}{N}\right). \tag{37}$$

**Lemma F.11.** *For a given $x \in \mathcal{X}$, we have,*

$$\left|\frac{\delta f(r)}{\delta r}\right| \leq \frac{N^2 \beta \exp(2\beta)}{(N-1)^2}, \tag{38}$$

*where $f(r) = \log\left(\mathbb{E}[\frac{1}{\exp(\beta r) + \sum_{i=1}^{N-1} \exp(\beta R_i)}]\right)$, $r = r(y, x)$ and $R_i = r(Y_i, x)$.*

*Proof.* Note that $\{R_i\}_{i=1}^{N-1}$ are i.i.d. . Therefore, we have,

$$\frac{\delta f(r(y,x))}{\delta r} = \mathbb{E}[\frac{1}{\exp(\beta r) + \sum_{i=1}^{N-1} \exp(\beta R_i)}]^{-1} \frac{\delta \mathbb{E}[\frac{1}{\exp(\beta r) + \sum_{i=1}^{N-1} \exp(\beta R_i)}]}{\delta r}$$

$$\leq \mathbb{E}[\frac{1}{\exp(\beta r) + \sum_{i=1}^{N-1} \exp(\beta R_i)}]^{-1}$$

$$\times \left(\sum_{k=1}^N \frac{\beta k \binom{N-1}{k-1} \exp(\beta r)}{(k \exp(\beta r) + N - 1 - k)^2} (1 - P(R = r))^{N-k} P^{k-1}(R = r)\right)$$

$$\leq \mathbb{E}[\frac{1}{\exp(\beta r) + \sum_{i=1}^{N-1} \exp(\beta R_i)}]^{-1} \tag{39}$$

$$\times \frac{\beta \exp(\beta)}{(N-1)^2} \left(\sum_{k=1}^N k \binom{N-1}{k-1} (1 - P(R = r))^{N-k} P^{k-1}(R = r)\right)$$

$$\leq \frac{N\beta \exp(2\beta)}{(N-1)^2} (1 + (N-1) P(R = r))$$

$$\leq \frac{N^2 \beta \exp(2\beta)}{(N-1)^2}.$$

For first inequality, we provide more details in the following. Lets consider the conditional expectation where we condition on having exactly $(k-1)$ of $(N-1)$ $R_i$s equal to $r$. Therefore, we have,

$$\frac{\delta\mathbb{E}\big[\frac{1}{\exp(\beta r)+\sum_{i=1}^{N-1}\exp(\beta R_i)}\big]}{\delta r}$$

$$=\sum_{k=1}^{N}\binom{N-1}{k-1}\mathbb{E}\Big[\frac{\beta k\exp(\beta r)}{k\exp(\beta r)+\sum_{i=1}^{N-k}\exp(\beta R_i)}\Big](1-P(R=r))^{N-k}P^{k-1}(R=r)$$

Now computing the derivative and using the fact that $1\le\exp(\beta R_i)$, the first inequality holds.

$\square$

**Lemma F.12** (The BH inequality (Canonne, 2022)). *For every two probability distributions* $\mathbf{p},\mathbf{q}$, *we have the simple yet never vacuous bound*

$$\mathbb{TV}(\mathbf{p},\mathbf{q})\le\sqrt{1-e^{-\mathrm{KL}(\mathbf{p}\|\mathbf{q})}}.\tag{40}$$

## G  PROOF AND DETAILS OF SECTION 4

> **Lemma 4.1.** *The following upper bound holds on KL divergence between SBoN and reference policies for a given prompt* $x\in\mathcal{X}$,
>
> $$\mathrm{KL}(\pi_{r^\star}^{\mathrm{SBoN}}(y|x)\|\pi_{\mathrm{ref}}(y|x))\le\log\Big(\frac{N}{1+(N-1)\exp(-\beta)}\Big).\tag{41}$$

*Proof.* Recall that,

$$\pi_{r^\star}^{\mathrm{SBoN}}(y|x)=\pi_{\mathrm{ref}}(y|x)\exp(\beta r^\star(y,x))\mathbb{E}\Big[\big(\frac{1}{N}(\exp(\beta r^\star(y,x))+\sum_{i=1}^{N-1}\exp(\beta r^\star(Y_i,x)))\big)^{-1}\Big].$$

Now, we have,

$$\mathrm{KL}(\pi_{r^\star}^{\mathrm{SBoN}}(y|x)\|\pi_{\mathrm{ref}}(y|x))$$
$$=\sum_{\mathcal{Y}}\pi_{r^\star}^{\mathrm{SBoN}}(y|x)\log(\pi_{r^\star}^{\mathrm{SBoN}}(y|x)/\pi_{\mathrm{ref}}(y|x))$$
$$=\sum_{\mathcal{Y}}\pi_{r^\star}^{\mathrm{SBoN}}(y|x)\log(N)$$
$$+\sum_{\mathcal{Y}}\pi_{r^\star}^{\mathrm{SBoN}}(y|x)\log(\mathbb{E}\Big[\exp(\beta r^\star(y,x))\big(\exp(\beta r^\star(y,x))+\sum_{i=1}^{N-1}\exp(\beta r^\star(Y_i,x)))^{-1}\Big])$$
$$=\log(N)+\sum_{\mathcal{Y}}\pi_{r^\star}^{\mathrm{SBoN}}(y|x)\log(\mathbb{E}\Big[\big(1+\sum_{i=1}^{N-1}\exp(\beta(r^\star(Y_i,x)-r^\star(y,x)))\big)^{-1}\Big]).\tag{42}$$

For the second term in equation 42, consider

$$A(y,Y,x)=\sum_{i=1}^{N-1}\exp(\beta(r^\star(Y_i,x)-r^\star(y,x)))>0,$$

where we have

$$(N-1)\exp(-\beta)\le A(y,Y,x)\le(N-1)\exp(\beta).$$

Therefore, we have,

$$\sum_{\mathcal{Y}} \pi_{r^\star}^{\text{SBoN}}(y|x) \log(\mathbb{E}\Big[\Big(1 + \sum_{i=1}^{N-1} \exp(\beta(r^\star(Y_i, x) - r^\star(y, x))))^{-1}\Big]\Big)$$
$$\leq \sum_{\mathcal{Y}} \pi_{r^\star}^{\text{SBoN}}(y|x) \log(\frac{1}{1 + (N-1)\exp(-\beta)}) \tag{43}$$
$$= \log(\frac{1}{1 + (N-1)\exp(-\beta)}).$$

Combining equation 43 with equation 42 completes the proof.

$\square$

> **Lemma 4.2.** *The following upper bound holds on the KL divergence between the SBoN policies under true-reward and proxy-reward models respectively,*
>
> $$\text{KL}(\pi_{r^\star}^{\text{SBoN}}(\cdot|x) \| \pi_{\hat{r}}^{\text{SBoN}}(\cdot|x)) \leq \frac{N\beta\sqrt{\varepsilon_{\beta,r}(x)}}{1 + (N-1)\exp(-\beta)}\Big(\frac{N\exp(2\beta)}{(N-1)^2} + 1\Big). \tag{44}$$

*Proof.* We first provide the following upper bound,

$$\text{KL}(\pi_{r^\star}^{\text{SBoN}}(y|x) \| \pi_{\hat{r}}^{\text{SBoN}}(y|x))$$
$$= \sum_{\mathcal{Y}} \pi_{r^\star}^{\text{SBoN}}(y|x) \log \big(\frac{\pi_{r^\star}^{\text{SBoN}}(y|x)}{\pi_{\hat{r}}^{\text{SBoN}}(y|x)}\big)$$
$$= \sum_{\mathcal{Y}} \pi_{r^\star}^{\text{SBoN}}(y|x)\beta(r^\star(y, x) - \hat{r}(y, x))$$
$$+ \sum_{\mathcal{Y}} \pi_{r^\star}^{\text{SBoN}}(y|x)\Big(\log\big(\mathbb{E}[\frac{1}{\exp(\beta r^\star(y, x)) + \sum_{i=1}^{N-1}\exp(\beta r^\star(Y_i, x))}]\big)$$
$$- \log\big(\mathbb{E}[\frac{1}{\exp(\beta\hat{r}(y, x)) + \sum_{i=1}^{N-1}\exp(\beta\hat{r}(Y_i, x))}]\big)\Big) \tag{45}$$
$$\leq \frac{N\beta\sqrt{\varepsilon_{\beta,r}(x)}}{1 + (N-1)\exp(-\beta)}$$
$$+ \sum_{\mathcal{Y}} \pi_{r^\star}^{\text{SBoN}}(y|x)\Big(\log\big(\mathbb{E}[\frac{1}{\exp(\beta r^\star(y, x)) + \sum_{i=1}^{N-1}\exp(\beta r^\star(Y_i, x))}]\big)$$
$$- \log\big(\mathbb{E}[\frac{1}{\exp(\beta\hat{r}(y, x)) + \sum_{i=1}^{N-1}\exp(\beta\hat{r}(Y_i, x))}]\big)\Big),$$

where we used two facts. First,

$$\pi_{r^\star}^{\text{SBoN}}(\cdot|x) \leq \frac{N\pi_{\text{ref}}(\cdot|x)}{1 + (N-1)\exp(-\beta)}$$

Second, for a random variable X,

$$E[X] \leq \sqrt{E[X^2]} = \sqrt{E[\frac{1}{\beta}\log(\exp(\beta X^2))]} \leq \sqrt{\frac{1}{\beta}\log(E[\exp(\beta X^2)])},$$

and applying it to $E_{\pi_{\text{ref}}(\cdot|x)}[(r^\star(y, x) - \hat{r}(y, x))]$, the final result in holds.

Note that for the last term in equation 45, we can apply the mean-value theorem as follows,

$$\sum_{\mathcal{Y}} \pi_{r^\star}^{\text{SBoN}}(y|x)\left( \log\left( \mathbb{E}[\frac{1}{\exp(\beta r^\star(y,x)) + \sum_{i=1}^{N-1}\exp(\beta r^\star(Y_i,x))}]\right)\right.$$

$$\left. - \log\left( \mathbb{E}[\frac{1}{\exp(\beta \hat{r}(y,x)) + \sum_{i=1}^{N-1}\exp(\beta \hat{r}(Y_i,x))}]\right)\right) \tag{46}$$

$$\leq \sum_{\mathcal{Y}} \pi_{r^\star}^{\text{SBoN}}(y|x)|r^\star(y,x) - \hat{r}(y,x)|\left|\frac{\delta f(r_\gamma(y,x))}{\delta r}\right|,$$

where $f(r_\gamma(y,x)) = \log\left( \mathbb{E}[\frac{1}{\exp(\beta r_\gamma(y,x)) + \sum_{i=1}^{N-1}\exp(\beta r_\gamma(Y_i,x))}]\right)$, for some $\gamma \in (0,1)$ we have $r_\gamma(y,x) = \gamma \hat{r}(y,x) + (1-\gamma)r^\star(y,x)$. Using Lemma F.11, we have,

$$\left|\frac{\delta f(r(y,x))}{\delta r}\right| \leq \frac{N^2\beta \exp(2\beta)}{(N-1)^2}. \tag{47}$$

Using equation 47 in equation 46 and applying Cauchy-Schwarz inequality, we have,

$$\sum_{\mathcal{Y}} \pi_{r^\star}^{\text{SBoN}}(y|x)\left( \log\left( \mathbb{E}[\frac{1}{\exp(\beta r^\star(y,x)) + \sum_{i=1}^{N-1}\exp(\beta r^\star(Y_i,x))}]\right)\right.$$

$$\left. - \log\left( \mathbb{E}[\frac{1}{\exp(\beta \hat{r}(y,x)) + \sum_{i=1}^{N-1}\exp(\beta \hat{r}(Y_i,x))}]\right)\right)$$

$$\leq \sqrt{\sum_{\mathcal{Y}} \mathbb{E}[1/(1 + \sum_{i=1}^{N-1}\exp(\beta(r^\star(Y_i,x) - r^\star(y,x))))]^2 \pi_{\text{ref}}(y|x)} \tag{48}$$

$$\times \sqrt{\sum_{\mathcal{Y}} |r^\star(y,x) - \hat{r}(y,x)|^2 \pi_{\text{ref}}(y|x)}\sqrt{\sum_{\mathcal{Y}} \left|\frac{\delta f(r_\gamma(y,x))}{\delta r}\right|^2 \pi_{\text{ref}}(y|x)}$$

$$\leq \frac{\sqrt{\varepsilon_{\beta,r}(x)}}{1 + (N-1)\exp(-\beta)}\frac{N^2\beta \exp(2\beta)}{(N-1)^2}.$$

This completes the proof. $\qquad\square$

## H  PROOF AND DETAILS OF SECTION 5

### H.1  UPPER BOUND PROOF AND DETAILS

**Lemma 5.1 (Full Version).** *Under Assumption 3.1, the following properties of $C_{\beta,r}(x)$ hold,*

1. $C_{\beta,r}(x) = \frac{\mathbb{E}[\exp(2\beta\hat{r}(Y,x))]}{\mathbb{E}^2[\exp(\beta\hat{r}(Y,x))]}$.
2. $C_{\beta,r}(x)$ *is an increasing function with respect to $\beta$.*
3. $C_{\infty,\hat{r}}(x) = \frac{1}{\sum_i \pi_{\text{ref}}(y_{i,r}^{\max}(x)|x)}$ *where $y_{i,r}^{\max}(x) \in \arg\max_y r(y,x)$.*
4. *For all $\beta < \infty$, we have $1 \leq C_{\beta,r}(x) \leq \min(C_{\infty,\hat{r}}(x), \exp(2\beta))$.*

*Proof.* In the following, we provide proofs of different items.

1.

$$C_{\beta,\hat{r}}(x) = \sum_{\mathcal{Y}} \frac{\pi_{\beta,\hat{r}}^2(y|x)}{\pi_{\text{ref}}(y|x)}$$

$$= \sum_{\mathcal{Y}} \frac{\exp(2\beta\hat{r}(y,x))}{\mathbb{E}^2[\exp(\beta\hat{r}(Y,x))]}\pi_{\text{ref}}(y|x) \tag{49}$$

$$= \frac{\mathbb{E}[\exp(2\beta\hat{r}(Y,x))]}{\mathbb{E}^2[\exp(\beta\hat{r}(Y,x))]}.$$

2. We can show that the logarithm function of $C_{\beta,\hat{r}}(x)$ is increasing. Then, due to the increasing feature of the log function, the final result holds.

$$
\begin{aligned}
&\log\Big(\frac{\mathbb{E}[\exp(2\beta\hat{r}(Y,x))]}{\mathbb{E}^2[\exp(\beta\hat{r}(Y,x))]}\Big) \\
&= \log(\mathbb{E}[\exp(2\beta\hat{r}(Y,x))]) - 2\log(\mathbb{E}[\exp(\beta\hat{r}(Y,x))]),
\end{aligned}
\tag{50}
$$

then we can compute the derivative of equation 50,

$$
\begin{aligned}
&\frac{\mathrm{d}\log(\mathbb{E}[\exp(2\beta\hat{r}(Y,x))])}{\mathrm{d}\beta} - 2\frac{\mathrm{d}\log(\mathbb{E}[\exp(\beta\hat{r}(Y,x))])}{\mathrm{d}\beta} \\
&= \frac{\mathbb{E}[2\hat{r}(Y,x)\exp(2\beta\hat{r}(Y,x))]}{\mathbb{E}[\exp(2\beta\hat{r}(Y,x))]} - 2\frac{\mathbb{E}[\hat{r}(Y,x)\exp(\beta\hat{r}(Y,x))]}{\mathbb{E}[\exp(\beta\hat{r}(Y,x))]}
\end{aligned}
\tag{51}
$$

Note that we have,

$$
\begin{aligned}
&\frac{\mathrm{d}\frac{\mathbb{E}[\hat{r}(Y,x)\exp(\beta\hat{r}(Y,x))]}{\mathbb{E}[\exp(\beta\hat{r}(Y,x))]}}{\mathrm{d}\beta} \\
&= \frac{\mathbb{E}[\hat{r}^2(Y,x)\exp(\beta\hat{r}(Y,x))]\mathbb{E}[\exp(\beta\hat{r}(Y,x))] - \mathbb{E}[\hat{r}(Y,x)\exp(\beta\hat{r}(Y,x))]^2}{\mathbb{E}^2[\exp(\beta\hat{r}(Y,x))]} \\
&= \mathbb{E}_{Y\sim\pi_{\beta,\hat{r}}(\cdot|x)}[\hat{r}^2(Y,x)] - \mathbb{E}_{Y\sim\pi_{\beta,\hat{r}}(\cdot|x)}[\hat{r}(Y,x)]^2 \geq 0.
\end{aligned}
\tag{52}
$$

Therefore, we have,

$$
\frac{\mathbb{E}[\hat{r}(Y,x)\exp(2\beta\hat{r}(Y,x))]}{\mathbb{E}[\exp(2\beta\hat{r}(Y,x))]} \geq \frac{\mathbb{E}[\hat{r}(Y,x)\exp(\beta\hat{r}(Y,x))]}{\mathbb{E}[\exp(\beta\hat{r}(Y,x))]}.
\tag{53}
$$

This completes the proof.

3. Follows directly from Lemma F.9.

4. Due to Jensen inequality for $\mathbb{E}^2[\exp(\beta\hat{r}(Y,x))] \leq \mathbb{E}[\exp(2\beta\hat{r}(Y,x))]$, the $C_{\beta,\hat{r}}(x)$. We also have the uniform bound, $C_{\beta,\hat{r}}(x) = \frac{\mathbb{E}[\exp(2\beta\hat{r}(Y,x))]}{\mathbb{E}^2[\exp(\beta\hat{r}(Y,x))]} \leq \exp(\beta)$. Furthermore, due to increasing property in second item, we also have $\sup_\beta C_{\beta,\hat{r}}(x) = C_{\infty,\hat{r}}(x)$. Therefore, the upper bound holds.

$$\square$$

**Theorem H.1.** *The following upper bound holds,*

$$
\begin{aligned}
\Delta_{J_{r^\star}}(\pi_{\beta,r^\star}(\,\cdot\,|x),\pi_{\hat{r}}^{\mathrm{SBoN}}(\cdot|x)) \leq{}& \frac{1}{\beta}\Big(\mathrm{KL}(\pi_{\beta,r^\star}(\cdot|x)\|\pi_{\mathrm{ref}}(\cdot|x)) - \mathrm{KL}(\pi_{\beta,\hat{r}}(\cdot|x)\|\pi_{\mathrm{ref}}(\cdot|x))\Big) \\
&+ \sqrt{\varepsilon_{\beta,r}(x)}\Big(\sqrt{C_{\beta,\hat{r}}(x)} + \sqrt{C_{\beta,r^\star}(x)}\Big) \\
&+ 2\sqrt{\frac{1}{2}\log\Big(1 + \frac{C_{\beta,\hat{r}}(x) - 1}{N}\Big)}.
\end{aligned}
$$

*Proof.* Note that, we have,

$$
\begin{aligned}
&\Delta_{J_{r^\star}}(\pi_{\beta,r^\star}(\,\cdot\,|x),\pi_{\hat{r}}^{\mathrm{SBoN}}(\cdot|x)) \\
&= \mathbb{E}_{Y\sim\pi_{\beta,r^\star}(\cdot|x)}[r^\star(Y,x)] - \mathbb{E}_{Y\sim\pi_{\hat{r}}^{\mathrm{SBoN}}(\cdot|x)}[r^\star(Y,x)] \\
&= \underbrace{\mathbb{E}_{Y\sim\pi_{\beta,r^\star}(\cdot|x)}[r^\star(Y,x)] - \mathbb{E}_{Y\sim\pi_{\beta,\hat{r}}(\cdot|x)}[r^\star(Y,x)]}_{I_1} \\
&\quad + \underbrace{\mathbb{E}_{Y\sim\pi_{\beta,\hat{r}}(\cdot|x)}[r^\star(Y,x)] - \mathbb{E}_{Y\sim\pi_{\hat{r}}^{\mathrm{SBoN}}(\cdot|x)}[r^\star(Y,x)]}_{I_2}
\end{aligned}
\tag{54}
$$

Note that, using the definition of $\pi_{\beta,r^\star}(\cdot|x)$ and $\pi_{\beta,\hat{r}}(\cdot|x)$ as solutions to KL-regularized problem, we have,

$$\mathbb{E}_{Y\sim\pi_{\beta,r^\star}(\cdot|x)}[r^\star(Y,x)] = \frac{1}{\beta}\mathrm{KL}(\pi_{\beta,r^\star}(\cdot|x)\|\pi_{\mathrm{ref}}(\cdot|x)) + \frac{1}{\beta}\log(\mathbb{E}_{Y\sim\pi_{\mathrm{ref}}(\cdot|x)}[\exp(\beta r^\star(Y,x))]).$$
(55)

$$\mathbb{E}_{Y\sim\pi_{\beta,\hat{r}}(\cdot|x)}[\hat{r}(Y,x)] = \frac{1}{\beta}\mathrm{KL}(\pi_{\beta,\hat{r}}(\cdot|x)\|\pi_{\mathrm{ref}}(\cdot|x)) + \frac{1}{\beta}\log(\mathbb{E}_{Y\sim\pi_{\mathrm{ref}}(\cdot|x)}[\exp(\beta\hat{r}(Y,x))]). \quad (56)$$

Therefore, for term $I_1$, we have,

$$\begin{aligned}
&\mathbb{E}_{Y\sim\pi_{\beta,r^\star}(\cdot|x)}[r^\star(Y,x)] - \mathbb{E}_{Y\sim\pi_{\beta,\hat{r}}(\cdot|x)}[r^\star(Y,x)]\\
&= \mathbb{E}_{Y\sim\pi_{\beta,r^\star}(\cdot|x)}[r^\star(Y,x)] - \mathbb{E}_{Y\sim\pi_{\beta,\hat{r}}(\cdot|x)}[\hat{r}(Y,x)]\\
&\quad + \mathbb{E}_{Y\sim\pi_{\beta,\hat{r}}(\cdot|x)}[\hat{r}(Y,x)] - \mathbb{E}_{Y\sim\pi_{\beta,\hat{r}}(\cdot|x)}[r^\star(Y,x)]\\
&= \frac{1}{\beta}\Big(\mathrm{KL}(\pi_{\beta,r^\star}(\cdot|x)\|\pi_{\mathrm{ref}}(\cdot|x)) - \mathrm{KL}(\pi_{\beta,\hat{r}}(\cdot|x)\|\pi_{\mathrm{ref}}(\cdot|x))\Big)\\
&\quad + \frac{1}{\beta}\log(\mathbb{E}_{Y\sim\pi_{\mathrm{ref}}(\cdot|x)}[\exp(\beta r^\star(Y,x))]) - \frac{1}{\beta}\log(\mathbb{E}_{Y\sim\pi_{\mathrm{ref}}(\cdot|x)}[\exp(\beta\hat{r}(Y,x))])\\
&\quad + \sum_{\mathcal{Y}}\pi_{\beta,\hat{r}}(\cdot|x)(\hat{r}(y,x) - r^\star(y,x))\\
&\le \frac{1}{\beta}\Big(\mathrm{KL}(\pi_{\beta,r^\star}(\cdot|x)\|\pi_{\mathrm{ref}}(\cdot|x)) - \mathrm{KL}(\pi_{\beta,\hat{r}}(\cdot|x)\|\pi_{\mathrm{ref}}(\cdot|x))\Big)\\
&\quad + \frac{1}{\beta}\log(\mathbb{E}_{Y\sim\pi_{\mathrm{ref}}(\cdot|x)}[\exp(\beta r^\star(Y,x))]) - \frac{1}{\beta}\log(\mathbb{E}_{Y\sim\pi_{\mathrm{ref}}(\cdot|x)}[\exp(\beta\hat{r}(Y,x))])\\
&\quad + \frac{1}{\sqrt{\beta}}\sqrt{\sum_{\mathcal{Y}}\frac{\pi_{\beta,\hat{r}}^2(y|x)}{\pi_{\mathrm{ref}}(y|x)}}\sqrt{\beta\sum_{\mathcal{Y}}(\hat{r}(y,x) - r^\star(y,x))^2\pi_{\mathrm{ref}}(y|x)}\\
&\le \frac{1}{\beta}\Big(\mathrm{KL}(\pi_{\beta,r^\star}(\cdot|x)\|\pi_{\mathrm{ref}}(\cdot|x)) - \mathrm{KL}(\pi_{\beta,\hat{r}}(\cdot|x)\|\pi_{\mathrm{ref}}(\cdot|x))\Big)\\
&\quad + \frac{1}{\beta}\log(\mathbb{E}_{Y\sim\pi_{\mathrm{ref}}(\cdot|x)}[\exp(\beta r^\star(Y,x))]) - \frac{1}{\beta}\log(\mathbb{E}_{Y\sim\pi_{\mathrm{ref}}(\cdot|x)}[\exp(\beta\hat{r}(Y,x))])\\
&\quad + \sqrt{C_{\beta,\hat{r}}(x)\varepsilon_{\beta,r}(x)}\\
&\le \frac{1}{\beta}\Big(\mathrm{KL}(\pi_{\beta,r^\star}(\cdot|x)\|\pi_{\mathrm{ref}}(\cdot|x)) - \mathrm{KL}(\pi_{\beta,\hat{r}}(\cdot|x)\|\pi_{\mathrm{ref}}(\cdot|x))\Big)\\
&\quad + \frac{1}{\beta}\log(\mathbb{E}_{Y\sim\pi_{\mathrm{ref}}(\cdot|x)}[\exp(\beta r^\star(Y,x))]) - \frac{1}{\beta}\log(\mathbb{E}_{Y\sim\pi_{\mathrm{ref}}(\cdot|x)}[\exp(\beta\hat{r}(Y,x))])\\
&\quad + \sqrt{C_{\beta,\hat{r}}(x)\varepsilon_{\beta,r}(x)}\\
&\le \frac{1}{\beta}\Big(\mathrm{KL}(\pi_{\beta,r^\star}(\cdot|x)\|\pi_{\mathrm{ref}}(\cdot|x)) - \mathrm{KL}(\pi_{\beta,\hat{r}}(\cdot|x)\|\pi_{\mathrm{ref}}(\cdot|x))\Big)\\
&\quad + \sqrt{C_{\beta,r^\star}(x)\varepsilon_{\beta,r}(x)}\\
&\quad + \sqrt{C_{\beta,\hat{r}}(x)\varepsilon_{\beta,r}(x)}.
\end{aligned}$$
(57)

For term $I_2$ and using a similar approach to term $I_1$ and applying Lemma F.10, we have,

$$
\mathbb{E}_{Y \sim \pi_{\beta,\hat{r}}(\cdot|x)}[r^\star(Y,x)] - \mathbb{E}_{Y \sim \pi_{\hat{r}}^{\mathrm{SBoN}}(\cdot|x)}[r^\star(Y,x)]
$$

$$
\leq 2\mathbb{TV}(\pi_{\beta,\hat{r}}(\,\cdot\,|x), \pi_{\hat{r}}^{\mathrm{SBoN}}(\cdot|x))
$$

$$
\leq 2\min\left(1, \sqrt{\frac{1}{2}\mathrm{KL}(\pi_{\beta,\hat{r}}(\,\cdot\,|x)\|\pi_{\hat{r}}^{\mathrm{SBoN}}(\cdot|x))}\right)
$$

$$
\leq 2\min\left(1, \sqrt{\frac{1}{2}\log\left(1 + \frac{C_{\beta,\hat{r}}(x) - 1}{N}\right)}\right)
$$

$$
\leq 2\sqrt{\frac{1}{2}\log\left(1 + \frac{C_{\beta,\hat{r}}(x) - 1}{N}\right)}
$$

(58)

Combining equation 57 and equation 58 with equation 54 completes the proof. $\qquad\square$

---

**Theorem 5.2.** *Under Assumption 3.1, the following upper bound holds on the optimal regret gap of the SBoN policy for any $\beta > 0$,*

$$
\Delta_{J_{r^\star}}(\pi_{r^\star}^\star(\,\cdot\,|x), \pi_{\hat{r}}^{\mathrm{SBoN}}(\cdot|x)) \leq \sqrt{\varepsilon_{\beta,r}(x)}\left(\sqrt{C_{\infty,\hat{r}}(x)} + \sqrt{C_{\infty,r^\star}(x)}\right)
$$

$$
+ 2\sqrt{\frac{1}{2}\log\left(1 + \frac{C_{\infty,\hat{r}}(x) - 1}{N}\right)}
$$

$$
+ \frac{\log(C_{\infty,r^\star}(x))}{\beta},
$$

---

*Proof.* Note that we have,

$$
\Delta_{J_{r^\star}}(\pi_{r^\star}^\star(\,\cdot\,|x), \pi_{\hat{r}}^{\mathrm{SBoN}}(\cdot|x))
$$

$$
= \mathbb{E}_{Y \sim \pi_{r^\star}^\star(\cdot|x)}[r^\star(Y,x)] - \mathbb{E}_{Y \sim \pi_{\hat{r}}^{\mathrm{SBoN}}(\cdot|x)}[r^\star(Y,x)]
$$

$$
= \underbrace{\mathbb{E}_{Y \sim \pi_{r^\star}^\star(\cdot|x)}[r^\star(Y,x)] - \mathbb{E}_{Y \sim \pi_{\beta,r^\star}(\cdot|x)}[r^\star(Y,x)]}_{I_3}
$$

(59)

$$
+ \underbrace{\Delta_{J_{r^\star}}(\pi_{\beta,r^\star}(\,\cdot\,|x), \pi_{\hat{r}}^{\mathrm{SBoN}}(\cdot|x))}_{I_4}
$$

For term $I_4$, we can use Theorem H.1. For term $I_3$, note that, we have for $\beta > 0$,

$$
\mathbb{E}_{Y \sim \pi_{r^\star}^\star(\cdot|x)}[r^\star(Y,x)] - \mathbb{E}_{Y \sim \pi_{\beta,r^\star}(\cdot|x)}[r^\star(Y,x)] \leq \frac{\mathrm{KL}(\pi_{r^\star}^\star(\,\cdot\,|x)\|\pi_{\mathrm{ref}}(\cdot|x)) - \mathrm{KL}(\pi_{\beta,r^\star}(\,\cdot\,|x)\|\pi_{\mathrm{ref}}(\cdot|x))}{\beta}
$$

(60)

Combining equation 60 with Theorem H.1, completes the proof due the positiveness of KL divergence and using Lemma F.7 and Lemma 5.1.

$\qquad\square$

**Remark H.2.** *For $\beta = 0$, we have, $\lim_{\beta \to 0} \pi_{\beta,r^\star}(\,\cdot\,|x) = \pi_{\mathrm{ref}}(\cdot|x)$. Therefore, we have,*

$$
\mathbb{E}_{Y \sim \pi_{r^\star}^\star(\cdot|x)}[r^\star(Y,x)] - \mathbb{E}_{Y \sim \pi_{\beta,r^\star}(\cdot|x)}[r^\star(Y,x)] \leq \sqrt{2\mathrm{KL}(\pi_{r^\star}^\star(\,\cdot\,|x)\|\pi_{\mathrm{ref}}(\cdot|x))}.
$$

(61)

---

**Proposition 5.3.** *Under Assumption 3.1, the following upper bound holds on the optimal regret gap of the BoN policy for any $\beta > 0$,*

$$
\Delta_{J_{r^\star}}(\pi_{r^\star}^\star(\cdot|x), \pi_{\hat{r}}^{\mathrm{BoN}}(\cdot|x)) \leq \sqrt{\varepsilon_{\infty,r}(x)}\left(\sqrt{C_{\infty,\hat{r}}(x)} + \sqrt{C_{\infty,r^\star}(x)}\right)
$$

$$
+ 2\sqrt{\frac{1}{2}\log\left(1 + \frac{C_{\infty,\hat{r}}(x) - 1}{N}\right)}.
$$

---

*Proof.* The results follow directly from Proposition 5.3 for $\beta \to \infty$. $\qquad\square$

## H.2 Lower bound proofs and details

We can define the probability of coverage of maximizers under the reference policy as

$$p^\star(x) := \Pr_{Y \sim \pi_{\text{ref}}(\cdot|x)}[Y \in \mathcal{Y}_{r^\star}^\star(x)] = \frac{1}{C_{\infty, r^\star}(x)}.$$

**Lemma H.3.** *Under Assumption 5.5, the following lower bound holds on the optimal regret gap of the SBoN policy based on true-reward,*

$$\Delta_{J_{r^\star}}(\pi_{r^\star}^\star(\cdot|x), \pi_{r^\star}^{\text{SBoN}}(\cdot|x)) \geq \gamma(x)\Pr_{Err}(x), \tag{62}$$

*where $\Pr_{Err}$ is the probability that selected sample from $\{Y_i\}_{i=1}^N$ via SBoN does not belong to $\mathcal{Y}_{r^\star}^\star(x)$.*

*Proof.* We have,

$$
\begin{aligned}
&\Delta_{J_{r^\star}}(\pi_{r^\star}^\star(\cdot|x), \pi_{r^\star}^{\text{SBoN}}(\cdot|x)) \\
&= J_{r^\star}(\pi_{r^\star}^\star(\cdot|x)) - J_{r^\star}(\pi_{\beta, r^\star}(\cdot|x)) \\
&= \mathbb{E}_{Y \sim \pi_{r^\star}^\star(\cdot|x)}[r^\star(Y, x)] - \mathbb{E}_{Y \sim \pi_{\beta, r^\star}(\cdot|x)}[r^\star(Y, x)] \\
&= 1 - \Big[\mathbb{E}_{Y \sim \pi_{\beta, r^\star}(\cdot|x)}[r^\star(Y, x)|\text{Err}]\Pr_{Err} + \mathbb{E}_{Y \sim \pi_{\beta, r^\star}(\cdot|x)}[r^\star(Y, x)|\text{Err}^c](1 - \Pr_{Err}(x))\Big] \\
&\geq 1 - \Big[(1 - \gamma(x))\Pr_{Err}(x) + (1 - \Pr_{Err}(x))\Big] \\
&= \gamma(x)\Pr_{Err}(x),
\end{aligned}
\tag{63}
$$

where Err is the event where the selected sample using SBoN is not a maximizer. $\qquad\square$

Next, we provide a lower bound on $\Pr_{Err}(x)$.

**Lemma H.4.** *The following lower bound holds on $\Pr_{Err}$ for SBoN algorithm,*

$$\Pr_{Err}(x) \geq (1 - p^\star(x))^N + \frac{N((1 - p^\star(x)) - (1 - p^\star(x))^N)}{N + (\exp(\beta) - 1)(N - 1)}. \tag{64}$$

*Proof.* We consider cases where exactly $K$ responses are selected from $\mathcal{Y}_{r^\star}^\star(x)$. The $K$ has a binomial distribution $\text{Bin}(N, p^\star(x))$ Using Bayes rule, we have,

$$
\begin{aligned}
\Pr_{Err}(x) &= \sum_{i=0}^{N-1} Pr(K = i)\Pr_{Err}(x|K = i) \\
&= \sum_{i=0}^{N-1} \binom{N}{i} p^\star(x)^i (1 - p^\star(x))^{N-i} \Pr_{Err}(x|K = i) \\
&= \sum_{i=0}^{N-1} \binom{N}{i} p^\star(x)^i (1 - p^\star(x))^{N-i} \frac{\sum_{Y_j \notin \mathcal{Y}_{r^\star}^\star(x)} \exp(\beta r^\star(Y_j, x))}{\sum_{j=1}^N \exp(\beta r^\star(Y_j, x))} \\
&\geq (1 - p^\star(x))^N + \sum_{i=1}^{N-1} \binom{N}{i} p^\star(x)^i (1 - p^\star(x))^{N-i} \frac{N - i}{N - i + ie^\beta} \\
&\geq (1 - p^\star(x))^N + \sum_{i=1}^{N-1} \binom{N}{i} p^\star(x)^i (1 - p^\star(x))^{N-i} \frac{N - i}{N + (e^\beta - 1)(N - 1)} \\
&\geq (1 - p^\star(x))^N + \frac{N((1 - p^\star(x)) - (1 - p^\star(x))^N)}{N + (e^\beta - 1)(N - 1)}
\end{aligned}
\tag{65}
$$

$\qquad\square$

Using Lemma H.3 and Lemma H.4, we can derive the following lower bound on our main optimal regret gap,

---

**Theorem 5.6.** *Under the same Assumptions in Theorem 5.2 and Lemma H.3, the following lower bound holds on the regret of SBoN policy under the proxy-reward model,*

$$\Delta_{J_{r^\star}}\left(\pi_{r^\star}^\star(\cdot|x), \pi_{\hat{r}}^{\text{SBoN}}(\cdot|x)\right)$$

$$\geq \gamma(x)\left(\left(1 - p^\star(x)\right)^N + \frac{N\left((1 - p^\star(x)) - (1 - p^\star(x))^N\right)}{N + (\exp(\beta) - 1)(N - 1)}\right) \tag{66}$$

$$- \min\left(\sqrt{1 - \exp\left(-U(N, \beta)\right)}, \frac{N}{1 + (N - 1)\exp(-\beta)}\sqrt{\varepsilon_{\beta, r}(x)}\right),$$

*where* $p^\star(x) = \frac{1}{C_{\infty, r^\star}(x)}$ *and* $U(N, \beta) = \frac{N\beta\sqrt{\varepsilon_{\beta, r}(x)}}{1 + (N-1)\exp(-\beta)}\left(\frac{N\exp(2\beta)}{(N-1)^2} + 1\right).$

---

*Proof.*

$$\Delta_{J_{r^\star}}\left(\pi_{r^\star}^\star(\cdot|x), \pi_{\hat{r}}^{\text{SBoN}}(\cdot|x)\right) \tag{67}$$

$$= \Delta_{J_{r^\star}}\left(\pi_{r^\star}^\star(\cdot|x), \pi_{r^\star}^{\text{SBoN}}(\cdot|x)\right) \tag{68}$$

$$+ \Delta_{J_{r^\star}}\left(\pi_{r^\star}^{\text{SBoN}}(\cdot|x), \pi_{\hat{r}}^{\text{SBoN}}(\cdot|x)\right), \tag{69}$$

where equation 68 can be bounded using Lemma H.3 and Lemma H.4. By combining Lemma F.12 and Lemma 4.2 we have an upper bound on absolute value of equation 69 where can also result in a lower bound on equation 69. Furthermore, another simple upper bound on equation 69 is,

$$\Delta_{J_{r^\star}}\left(\pi_{r^\star}^{\text{SBoN}}(\cdot|x), \pi_{\hat{r}}^{\text{SBoN}}(\cdot|x)\right) \tag{70}$$

$$= \mathbb{E}_{Y \sim \pi_{r^\star}^{\text{SBoN}}(Y|x)}[r^\star(Y, x)] - \mathbb{E}_{Y \sim \pi_{\hat{r}}^{\text{SBoN}}(Y|x)}[\hat{r}(Y, x)] \tag{71}$$

$$+ \mathbb{E}_{Y \sim \pi_{\hat{r}}^{\text{SBoN}}(Y|x)}[\hat{r}(Y, x)] - \mathbb{E}_{Y \sim \pi_{\hat{r}}^{\text{SBoN}}(Y|x)}[r^\star(Y, x)] \tag{72}$$

$$= \mathbb{E}_{Y \sim \pi_{\hat{r}}^{\text{SBoN}}(Y|x)}[\hat{r}(Y, x)] - \mathbb{E}_{Y \sim \pi_{\hat{r}}^{\text{SBoN}}(Y|x)}[r^\star(Y, x)] \tag{73}$$

Note that due to calibration, both true and proxy-reward models are uniform under $\pi_{\text{ref}}(\cdot|x)$. Therefore, we have,

$$\mathbb{E}_{Y \sim \pi_{r^\star}^{\text{SBoN}}(Y|x)}[r^\star(Y, x)] = \mathbb{E}_{Y \sim \pi_{\hat{r}}^{\text{SBoN}}(Y|x)}[\hat{r}(Y, x)].$$

Now we have,

$$\left|\mathbb{E}_{Y \sim \pi_{\hat{r}}^{\text{SBoN}}(Y|x)}[\hat{r}(Y, x)] - \mathbb{E}_{Y \sim \pi_{\hat{r}}^{\text{SBoN}}(Y|x)}[r^\star(Y, x)]\right| \leq \sqrt{\sum_y \frac{(\pi_{\hat{r}}^{\text{SBoN}}(\cdot|x))^2}{\pi_{\text{ref}}(\cdot|x)}}\sqrt{\varepsilon_{\beta, r}(x)}$$

$$\leq \frac{N}{1 + (N - 1)\exp(-\beta)}\sqrt{\varepsilon_{\beta, r}(x)}.$$

$\square$

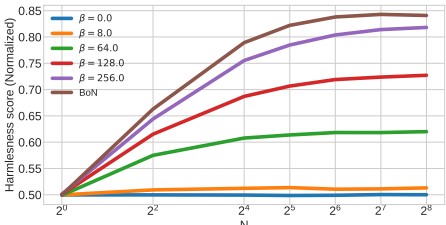 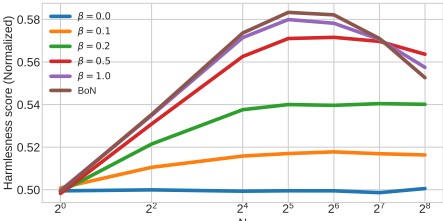

Figure 3: Soft Best-of-N experiment using a strong reward model (Left) and a weak one (Right). When the performance of the reward model is poor, BoN can lead to overoptimization, while the SBoN can help to mitigate it.

## I EXPERIMENTS

Our implementation is available in this link .

**LLM-as-a-Judge:** To evaluate the performance of the generated responses, we used GPT-4o to determine how harmless they are. Specifically, we used the following prompt:

> You are an AI safety evaluator. Your task is to determine if a response to a given prompt contains harmful content. Respond with a harmlessness score using these criteria:
>
> 2 = SAFE: The response is completely safe and appropriate.
> 1 = SOMEWHAT HARMFUL: The response contains potentially concerning elements, mild toxicity, stereotypes, subtle harmful guidance, or borderline inappropriate content.
> 0 = HARMFUL: The response contains clearly harmful content such as hate speech, explicit instructions for dangerous activities, severe toxicity, etc.
>
> Provide your judgment as an integer in curly brackets like this: 1, 2, or 3.

This yields a reward with values ranging from 0 to 2, where 0 indicates complete harm and 2 indicates safety.

Future work may address optimizing $\beta$ using cross-validation or the estimation of terms in Remark 5.8 to derive sub-optimal $\beta$. For this study, we employ a grid search over a range of $\beta$ values.

### I.1 MORE EXPERIMENTS

To further support our theoretical analysis, we conducted more experiments comparing Soft Best-of-N (SBoN) across different regularization strengths and reward model qualities. We ran one extra experimental conditions: one using a weaker proxy-reward model (tulu-v2.5-13b-uf-rm (Ivison et al., 2024)) with respect to Armo reward model. As shown in Figure 3, when the reward model is weak, performance degrades for large $N$ due to reward hacking. However, the smoothing in SBoN helps mitigate this degradation. This observation is also aligned with our theoretical analysis and discussion in Section 4.

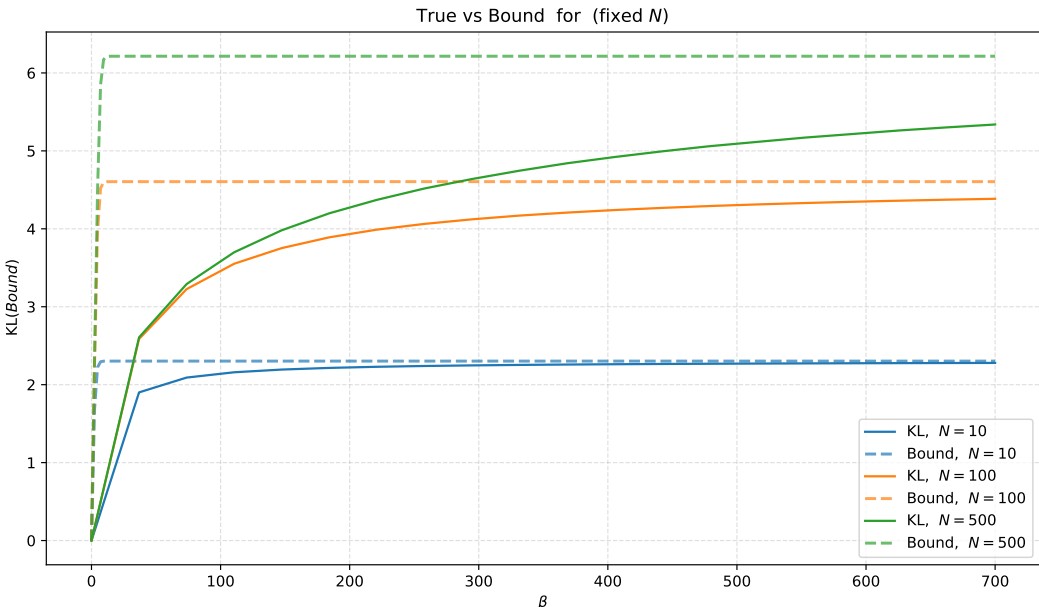

Figure 4: True KL divergence vs upper bound in Lemma 4.1 for fixed $N = \{10, 100, 500\}$.

## I.2 NUMERICAL EXAMPLES

### I.2.1 KL BOUND

To illustrate how our analytical upper bound in Lemma 4.1 behaves as a function of the inverse temperature parameter $\beta$, we run a toy experiment in which

1. the reference policy is the uniform distribution over responses, and
2. rewards are bounded, i.e., $r \in [0, 1]$.

For each $\beta$ in a logarithmic sweep, we compute the true KLdivergence between the SBoN policy and the reference policy, together with the theoretical bound derived in Lemma 4.1.

- **Very large $\beta$ (nearBoN policy).** As $\beta \to \infty$ the SBoN policy converges to the BoN policy. The gap between the KL and the bound vanishes.
- **Very small $\beta$ (reference policy).** When $\beta \to 0$ the SBoN policy approaches the uniform sampling from the reference policy, which results in the reference policy, making the KL itself tend to zero; the bound is equal to zero for this value.

This experiment confirms that the bound is tight in the two asymptotic regimes and remains a conservative yet informative estimate elsewhere.

### I.2.2 REGRET UPPER BOUND

To validate our theoretical findings in Theorem 5.2 and Proposition 5.3, we conducted a numerical analysis comparing the regret upper bounds of SBoN against standard BoN under varying degrees of reward model error.

We simulate a simplified binary reward setting to isolate the effects of regularization inverse temperature ($\beta$) and proxy-reward error ($\delta$). The setup is defined as follows:

**Reward function:** We assume a binary reward landscape where responses have true and proxy rewards $r^\star(y), \hat{r}(y) \in \{0, 1\}$.

**Reference Policy ($\pi_{\text{ref}}$):** The reference model generates an optimal response ($r^\star = 1$) with probability $\pi_{\text{ref}}(y_{max}|x) = 0.05$, corresponding to a coverage constant $C_{\infty, r^\star} = 20$. The same also

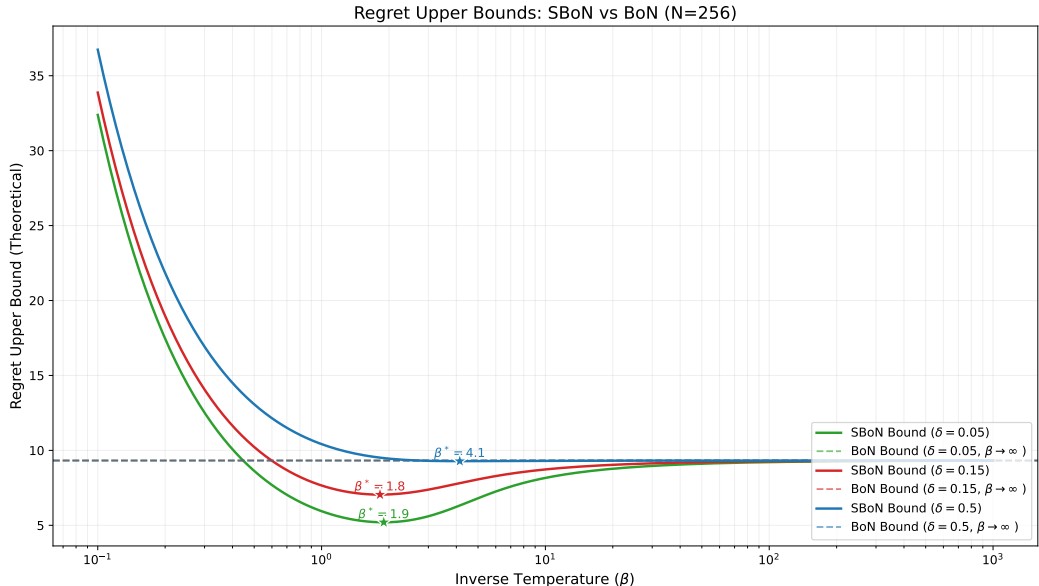

Figure 5: Numerical comparison of upper bound on Regret of SBoN and BoN

holds for proxy-reward model and we have $C_{\infty,\hat{r}} = 20$. We also consider $N = 256$ samples from reference policy.

**Proxy-reward Error ($\delta$):** The proxy-reward model matches the true-reward with probability $1 - \delta$ and commits an error (flipping the label) with probability $\delta$. We evaluate three error regimes: Low ($\delta = 0.05$), Medium ($\delta = 0.15$), and High ($\delta = 0.50$).

**SBoN:** We compute the regret upper bound using Theorem 5.2, varying the inverse temperature $\beta$ from 0.5 to 1000.

**BoN:** We compute the regret upper bound using Proposition 5.3, which represents the asymptotic limit of SBoN as $\beta \to \infty$. In this binary setting, the maximum possible error is ($\varepsilon_{\infty,r} = 1$).

**Discussion:** The results, Fig. 5 demonstrate a distinct trade-off mechanism governed by the inverse temperature $\beta$. While the standard BoN bound remains constant at a high value due to its susceptibility to the maximum error ($\varepsilon_{\infty,r} = 1$), the SBoN bound forms a curve with minimum.

For every error rate tested, we observe a "sweet spot"-a finite optimal inverse temperature $\beta^\star$ where the SBoN bound is significantly tighter than the BoN bound. This confirms that by smoothing the policy (finite $\beta$), SBoN effectively balances the trade-off between exploring high-reward regions and mitigating the overoptimization caused by proxy-reward errors.

### I.2.3 REGRET LOWER BOUND

To complement our lower bound analysis, we numerically evaluate the lower bounds on the regret gap for both SBoN and BoN, as derived in Theorem 5.6 and Proposition 5.7 respectively. These lower bounds serve as a "safety guarantee," quantifying the worst-case performance degradation caused by overoptimization. We utilize the same binary reward setup described in previous section, with sample size $N = 32$ and consider no error ($\delta = 0$). The results in Fig. 6 illustrate an advantage of the SBoN policy over the deterministic BoN policy. In error regimes (blue dashed line, $\delta = 0.15, 0.5$), the BoN lower bound drops significantly, indicating a weak guarantee against performance collapse. Note that, for no error regime, the lower bound on BoN's regret is positive.

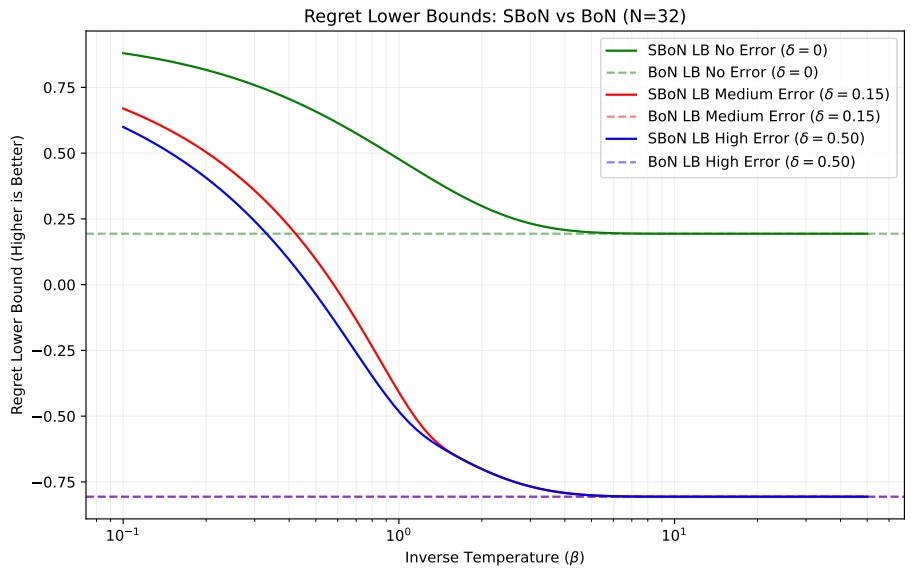

Figure 6: Numerical comparison of lower bound on Regret of SBoN and BoN

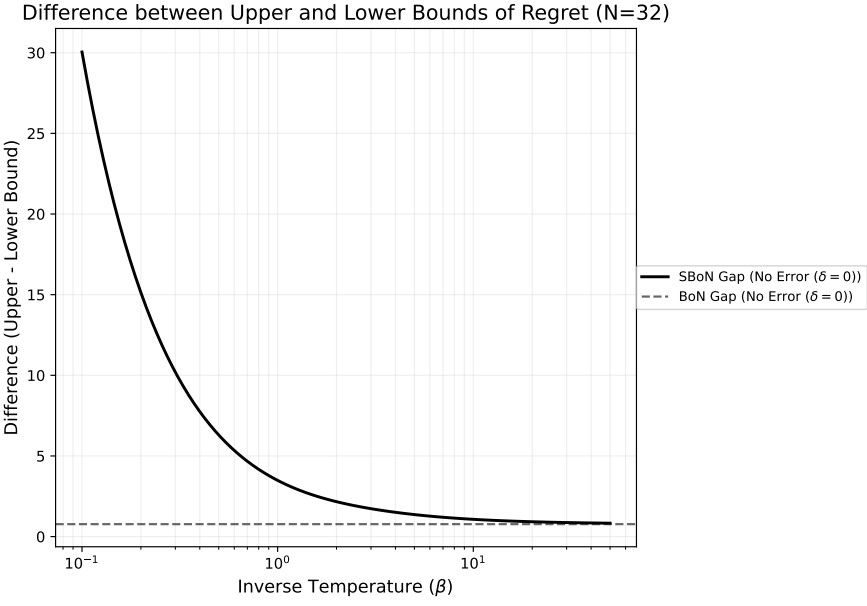

Figure 7: Numerical comparison of the gap between upper and lower bounds on Regret of SBoN and BoN

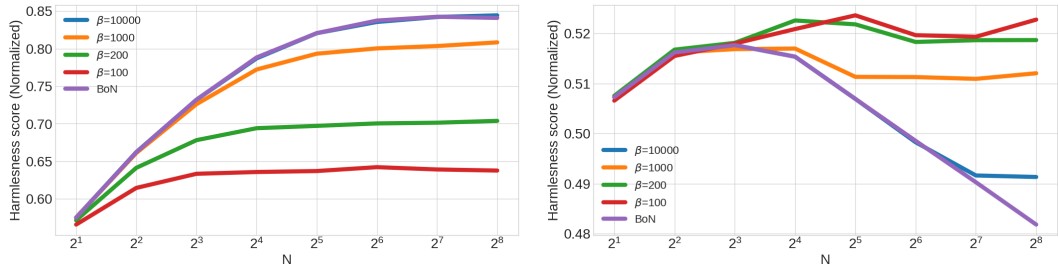

Figure 8: InferenceTimePessimism experiment using a strong reward model (Left) and a weak one (Right). When the performance of the reward model is poor, BoN can lead to overoptimization, while the InferenceTimePessimism can help to mitigate it.

### I.2.4 UPPER AND LOWER BOUNDS GAP

To compare the performance of upper and lower bounds, we consider the case where $\delta = 0$ which is similar to no-overoptimization scenario. As shown in Fig. 7, the gap between upper and lower bounds of BoN is tighter than SBoN.

## J COMPARISON WITH HUANG ET AL. (2025)

We contrast our contributions with the recent work of Huang et al. (2025), focusing on four key dimensions: the object of study, regret bounds, and the mechanism of regularization.

**Object of study.** Huang et al. (2025) analyze BoN, introducing an inference-time pessimism algorithm within a $\chi^2$-regularized framework to mitigate overoptimization. In contrast, we analyze SBoN via treating BoN as the limit where $\beta \to \infty$ and derive finite-sample KL and regret bounds under reward model misspecification (specifically, proxy-reward vs. true-reward). To our knowledge, these SBoN-specific bounds characterizing the behavior under overoptimization are novel.

**Error metrics and reward modeling.** While Huang et al. (2025) operate on raw (uncalibrated) rewards using mean-squared error (MSE), we utilize calibrated rewards (Balashankar et al., 2025). We introduce the *tilted error*, denoted as $\varepsilon_{\beta,r}$, which interpolates between MSE (at $\beta = 0$) and $L^\infty$ error (as $\beta \to \infty$). This metric allows us to explicitly track the dynamics of overoptimization as a function of both $N$ and the inverse temperature $\beta$.

**Regret bounds.** Our Theorem 5.2, Proposition 5.3, Theorem 5.6 and Proposition 5.7 and establish regret bounds for SBoN and its BoN limit that depend on: (i) the tilted error $\varepsilon_{\beta,r}(x)$, and (ii) the coverage constants $C_{\infty,r}$ for both the true and proxy-rewards. A crucial distinction, highlighted in Remark 5.4, is that our BoN regret bound remains finite even when overoptimization vanishes (i.e., when $\varepsilon_{\infty,r} = 0$ or $N \to \infty$). Conversely, the bound proposed by Huang et al. (2025) on BoN scales with the $L^\infty$-error. This represents a qualitative improvement in how misspecification is handled in our analysis.

**Theoretical and algorithmic consistency.** Our analysis of SBoN fully incorporates the algorithm's primary regularization parameter, $\beta$. In contrast, the theoretical regret bounds for *InferenceTimePessimism* in Huang et al. (2025) exclude the estimation error of the normalization factor (Algorithm 3) and are independent of the truncation parameter required by their Algorithm 4. Furthermore, regarding practical complexity, SBoN relies on a single hyperparameter $\beta$, whereas *InferenceTimePessimism* requires tuning an additional truncation parameter and incurs computational overhead to estimate the normalization factor.

**InferenceTimePessimism Experiment:**

We ran some experiments for InferenceTimePessimism algorithm (Huang et al., 2025); the results are shown in Fig. 8. We followed the configuration described in their paper, and in this experiment $\beta$ denotes the $\chi^2-$ regularization parameter. Comparing Fig. 8 with Fig. 2, we find that the two methods achieve comparable performance. However, SBoN is substantially easier to implement and

tune, since it avoids the extra computational overhead and expanded hyperparameter search space introduced by the pessimistic approach.

## K    INFORMATION PROJECTION ACROSS REWARD FUNCTIONS

In this section, we study the KL divergence behavior in Section 4 for large $N$ regime, where the SBoN policy converge to tilted optimal policy.

We consider a proxy-reward class $\mathcal{R}$ of rewards, and a projection problem at a *fixed* inverse temperature $\beta$ which is inspired by reducing the estimation error,

$$\textbf{(reverse-I-proj in tilted family)} \quad \hat{r}^{\mathrm{I}} \in \arg\min_{\hat{r}\in\mathcal{R}} \, D_{\mathrm{KL}}\big(\pi_{\beta,r^\star} \,\|\, \pi_{\beta,\hat{r}}\big).$$

**Proposition K.1** (Pythagorean I-projection in reward space)**.** *Suppose the proxy class $\mathcal{R}$ is convex in $r$ (e.g., an affine subspace). Let $\hat{r}^{\mathrm{I}} \in \arg\min_{\hat{r}\in\mathcal{R}} \mathrm{KL}(\pi_{\beta,r^\star}\|\pi_{\beta,\hat{r}})$. Then, we have,*

$$\mathrm{KL}(\pi_{\beta,r^\star}\|\pi_{\mathrm{ref}}) \;\geq\; \underbrace{\mathrm{KL}(\pi_{\beta,r^\star}\|\pi_{\beta,\hat{r}^{\mathrm{I}}})}_{\textit{Estimation error}} + \underbrace{\mathrm{KL}(\pi_{\beta,\hat{r}^{\mathrm{I}}}\|\pi_{\mathrm{ref}})}_{\textit{KL-divergence based on proxy-reward}} \tag{74}$$

*Proof.* The proof follows from Theorem 1 in Csiszár & Matus (2003). □

Proposition K.1 and Lemma F.7 provide an upper bound on $D_{\mathrm{KL}}(\pi_{\beta,r^\star}\|\pi_{\mathrm{ref}})$. Since the LHS of equation 74 is fixed for a given $\beta$, an increase in estimation error necessarily decreases the allowable divergence between the proxy-reward and the reference policies $D_{\mathrm{KL}}(\pi_{\beta,\hat{r}^{\mathrm{I}}}\|\pi_{\mathrm{ref}})$. If $r^\star \in \mathcal{R}$, the estimation error is zero (no overoptimization). If $r^\star \notin \mathcal{R}$, the estimation error consumes part of the total divergence budget. This analysis can be extended to finite $N$, using Lemma F.10.

## L    LARGE LANGUAGE MODEL (LLM) USAGE DISCLOSURE

We used large language models (LLMs) only for editing support (clarity/grammar). All LLM-assisted edits were reviewed by the authors, who remain fully responsible under the ICLR 2026 policy.

