# OpenReview forum: "Best-of-N through the Smoothing Lens: KL Divergence and Regret Analysis"
_ICLR.cc/2026/Conference — ICLR 2026 Poster_

### Official Review · Reviewer_vx3U · 2025-10-25

**Soundness:** 4
**Presentation:** 2
**Contribution:** 2
**Rating:** 4
**Confidence:** 4

**Summary:**

This paper studies Soft Best-of-N (SBoN), a smoothed variant of Best-of-N (BoN), from a theoretical perspective.

It proves (i) an explicit upper bound on the KL divergence between the SBoN policy and the reference policy (Lemma 4.1); (ii) an upper bound on the KL divergence between SBoN policies induced by true vs. proxy rewards as a function of (among others) the tilted error between proxy and true rewards (Lemma 4.2); (iii) an upper bound on true reward under proxy-reward SBoN (Theorem 4.3); and (iv) optimality-gap (regret) bounds for SBoN and, as a limit, BoN (Theorems 5.2, 5.3).

Empirically, SBoN mitigates (but does not eliminate) over-optimization when the proxy is weak (Fig. 1). Overall, the paper formally shows that smoothing can be a principled mitigation against reward over-optimization.

**Strengths:**

S1: The study of inference-time alignment is timely and rigorous, and the presented results target practically relevant and meaningful quantities. Best-of-N in commonly used in practice, so analyzing a smoothed version that could be safer against overoptimization is valuable.

S2: The mathematical setup is more convincing than previous related work because of the explicit reward calibration. I also appreciate that the authors considered the “tilted error” viewpoint since it gives a single knob that moves from average-case to worst-case sensitivity. It is a helpful way to reason about when smoothing should help, and I believe this setup is a great baseline for other papers in this field.

**Weaknesses:**

W1: The presentation can be greatly improved. In particular, I encourage the authors to motivate their setup. For example, they mention calibration in Line 49 without any context. The usefulness of calibration is not obvious until much later in Line 229. I also encourage them to be explicit in how calibration is implemented.

W2: While the theoretical results hold for BoN as well, they seem to be much looser than if BoN had been studied directly as in previous work. I understand why the bounds are loose because of the challenges analyzing the complex SBoN policy in (3). Hence, I believe readers would benefit from a clearer motivation for studying Soft BoN. This is especially true given that SBoN is a new method that is not deployed in practice.

W3:  I am not sure of the takeaway of this paper. This is due to two issues: (1) the obvious one is the lack of a principled way to choose the right temperature, as well as the limited experimental results, and (2) more importantly, I cannot make sense of what the results tell us. I would appreciate if the authors look at simple cases: how would smoothing help when the reward model misranks near the top only, for example?

Minor:
1. There are a few typos (missing spaces, spelling mistakes) scattered in the paper (lines 36, 46, 110...).
2. I would encourage moving the preview of the results in Tables 1 and 2 to the main text. I also liked Figure 2 so I can make sense of the different terms. I feel it is challenging to keep up with the notation in the main text. I would suggest looking at a different notation to differentiate the policies from each other.
3. At Line 35, fix the attribution of Best-of-N. It was popularized for LLMs by Stiennon et al.
4. Line 50 is confusing, what do you mean by "using some dataset"?

**Questions:**

Q1: Line 60: What do you mean by "scales proportionally" in line 60?

Q2: Line 273: what do you mean by KL regularized objective for SBoN? SBoN doesn't optimize that directly.

Q3: How useful are the upper bounds? How do you foresee other researchers operationalizing them to find or evaluate other strategies against reward hacking? I appreciate the toy example in Figure 4 for your first lemma and would suggest the same for your other results. Also, how would the upper bounds look like in your empirical experiment in Figure 1?

Q4: I understand finding lower bounds must be challenging, but are there settings where we could have a meaningful lower bound?

I am convinced by the authors' results. However, I believe the paper would benefit from a stronger motivation, as well as insights on the consequences of their theoretical results.

---

> ### Author Response · Authors · 2025-11-21
> **Response (1/2)**
>
> We thank the reviewer for the careful reading and constructive feedback. Below we address the main concerns and questions.
>
>
> > W2: While the theoretical results hold for BoN as well, they seem to be much looser than if BoN had been studied directly as in previous work. I understand why the bounds are loose because of the challenges analyzing the complex SBoN policy in (3). Hence, I believe readers would benefit from a clearer motivation for studying Soft BoN. This is especially true given that SBoN is a new method that is not deployed in practice.
>
> **R1:** We thank the reviewer for these insightful comments. We agree that the asymptotic analysis through SBoN yields a looser KL divergence bound compared to direct combinatorial approaches like Beirami et al. (2024) or Mroueh (2024), as we acknowledged in Section 4 (Eq 14 vs 11).
> However, regarding Regret bounds, we respectfully note that our results provide distinct advantages over prior work. For instance, unlike the bounds in Huang et al. (2025) which scale with the ​ error, our bounds remain finite even as the overoptimization error vanishes (Remark 5.4). If the reviewer has a specific reference in mind regarding tighter regret bounds (rather than KL), we would be grateful for the pointer to ensure our comparison is complete.
> Regarding the motivation for SBoN:
>
> 1. Theoretical Lens: SBoN provides a continuous "smoothing lens" (via $\\beta$) to analyze BoN. This allows us to analytically decompose the "overoptimization" phenomenon into a trade-off between estimation error and potential improvement (KL and regret bounds), showing analytically why an intermediate $\\beta$ often yields lower regret than $\\beta\\rightarrow \\infty$ pure BoN).
> 2. Overoptimization Mitigation Mechanism: As noted in our experiments (Figure 1), SBoN is not just a theoretical construct but a practical mitigation strategy. By tuning $\\beta$, we can act against Goodhart's Law in regimes where the proxy reward is weak, achieving performance that standard BoN cannot."
>
> > W3.1: I am not sure of the takeaway of this paper. This is due to two issues: (1) the obvious one is the lack of a principled way to choose the right temperature, as well as the limited experimental results, (2) more importantly,I cannot make sense of what the results tell us. I would appreciate if the authors look at simple cases: how would smoothing help when the reward model misranks near the top only, for example
>
> **R2:** We thank the reviewer for this insightful comment. We agree that understanding how to select $\\beta$ and the practical implications of our experiments are crucial for the paper's takeaways. We address these two concerns below:
>
> 1. On the principled choice of temperature ($\\beta$): We can try cross-validation which is the standard practical approach (as is common with hyperparameters). Our approach in Remark 5.8 can be applied for a data-driven approach of choosing $\beta$. However, due to errors of estimation, we preferred to choose $\beta$ via grid search.  Furthermore,
> * Theoretical Trade-off: As detailed in Remark 4.5 and Theorem 4.3, the choice of $\\beta$ governs the trade-off between maximizing the proxy reward and mitigating the "tilted error" (estimation error).
> * The "Sweet Spot": In Section 5.3 (Remark 5.8) and the numerical analysis in Figure 5 (Appendix I.4), we demonstrate that for any non-zero proxy error, there exists a finite optimal $\\beta^\\star$  that minimizes regret.
> * Practical Heuristic: Our theory suggests a principled heuristic: as the quality of the proxy reward decreases (higher estimation error), $\\beta$ must be lowered to increase smoothing. Conversely, as N increases, overoptimization risk rises, also necessitating a lower $\\beta$ (smoothing) compared to pure BoN (where $\\beta\\rightarrow \\infty$).
> 2. On the limited experimental results: We believe our experiments effectively validate the theoretical claims, but we acknowledge we can highlight them better. We have included substantial additional empirical evidence in Appendix I:
> * Robustness to Weak Proxies: Figure 1 and Figure 3 (Appendix) demonstrate that while standard BoN suffers from performance degradation (reward hacking) as N increases under weak reward models, SBoN with a finite $\\beta$ remains robust.
> * Comparison to Baselines: We compare SBoN against the "Inference-Time Pessimism" method (Huang et al., 2025\) in Figure 8 (Appendix J). The results show SBoN achieves similar performance while being significantly simpler to implement (tuning a single scalar $\\beta$ vs. truncation parameters and normalization factors).
> * Validation of Bounds: Figure 4, 5, 6 and 7 numerically validate our derived KL and regret bounds (lower, upper and their difference), confirming the theoretical gap between SBoN and BoN.

---

> ### Author Response · Authors · 2025-11-21
> **Response (2/2)**
>
> > W2: While the theoretical results hold for BoN as well, they seem to be much looser than if BoN had been studied directly as in previous work. I understand why the bounds are loose because of the challenges analyzing the complex SBoN policy in (3). Hence, I believe readers would benefit from a clearer motivation for studying Soft BoN. This is especially true given that SBoN is a new method that is not deployed in practice.
>
> **R3:** We thank the reviewer for these insightful comments. We agree that the asymptotic analysis through SBoN yields a looser KL divergence bound compared to direct combinatorial approaches like Beirami et al. (2024) or Mroueh (2024), as we acknowledged in Section 4 (Eq 14 vs 11).
> However, regarding Regret bounds, we respectfully note that our results provide distinct advantages over prior work. For instance, unlike the bounds in Huang et al. (2025) which scale with the ​ error, our bounds remain finite even as the overoptimization error vanishes (Remark 5.4). If the reviewer has a specific reference in mind regarding tighter regret bounds (rather than KL), we would be grateful for the pointer to ensure our comparison is complete.
> Regarding the motivation for SBoN:
>
> 1. Theoretical Lens: SBoN provides a continuous "smoothing lens" (via $\\beta$) to analyze BoN. This allows us to analytically decompose the "overoptimization" phenomenon into a trade-off between estimation error and potential improvement (KL and regret bounds), showing analytically why an intermediate $\\beta$ often yields lower regret than $\\beta\\rightarrow \\infty$ pure BoN).
> 2. Overoptimization Mitigation Mechanism: As noted in our experiments (Figure 1), SBoN is not just a theoretical construct but a practical mitigation strategy. By tuning $\\beta$, we can act against Goodhart's Law in regimes where the proxy reward is weak, achieving performance that standard BoN cannot."
>
> >**Q1:** Line 60: What do you mean by "scales proportionally" in line 60?
>
> **R4:**  Thank you for catching this ambiguity. On line 60 we intended to summarize the scaling behavior observed in Gao et al. (2023b) and Hilton et al. (2022), namely that under a calibrated true-reward model, the *improvement* in expected calibrated true reward over the reference policy behaves approximately like
>  $$O(KL(\\pi \\| \\pi\_{ref}))$$
> for both RL and BoN policies.
>
> To avoid confusion, we changed the  "scales proportionally" to “grows approximately proportionally to”.
>
> > Q2: Line 273: what do you mean by KL regularized objective for SBoN? SBoN doesn't optimize that directly.
>
> **R5:**  Thanks for pointing out this typo. It is fixed.
>
> > Q3: How useful are the upper bounds? How do you foresee other researchers operationalizing them to find or evaluate other strategies against reward hacking? I appreciate the toy example in Figure 4 for your first lemma and would suggest the same for your other results. Also, how would the upper bounds look like in your empirical experiment in Figure 1?
>
> **R6:** We thank the reviewer for the suggestion. The upper bounds (Theorem 4.3 and 5.2) are useful because they mathematically decouple the **estimation error** (from the proxy) from the **optimization gain**. Researchers can operationalize this to derive "scaling laws" for alignment: specifically, determining how to decrease $\\beta$ (increase smoothing) as $N$ grows to maintain optimal regret, rather than relying on fixed hyperparameters. Furthermore, we provide a numerical visualization of these bounds in **Figure 5 (Appendix I.4)**. The theoretical bounds in Figure 5 mirror the empirical trends in Figure 1: the BoN bound stagnates due to maximum error, while the SBoN bound forms a convex curve with a distinct "sweet spot" for $\\beta$ that minimizes regret.
>
> > Q4: I understand finding lower bounds must be challenging, but are there settings where we could have a meaningful lower bound?
>
> **R7:** Thanks for your helpful  comment. The lower bound results are provided in this general post….
>
> > I am convinced by the authors' results. However, I believe the paper would benefit from a stronger motivation, as well as insights on the consequences of their theoretical results.
>
> **R8:** We appreciate that the reviewers are convinced by our results and hope that our rebuttal addresses your concerns.
>
> ---
>
> **References:**
>
> * (Balashankar et al., 2024\) "InfAlign: Inference-aware language model alignment." *ICML 2025*.
> * (Beirami et al, 2025\) “Theoretical guarantees on the best-of-n alignment policy.” (ICML, 2024).
> * ( Mroueh et al, 2025\) “Information theoretic guarantees for policy alignment in large language models.” (TMLR, 2025\)
> * Huang et al. (2025): Huang, Audrey, et al. "Is Best-of-N the Best of Them? Coverage, Scaling, and Optimality in Inference-Time Alignment." *ICML 2025*.

---

> > ### Comment · Reviewer_vx3U · 2025-11-23
> >
> > Thanks for the reply! Your answers are very helpful.
> >
> > My only remaining recommendation is to improve the presentation of the results by simplifying the notation, being clearer about the takeaways, as well as streamlining your setup as I mention in W1. This will make your paper much more accessible to the larger machine learning community.

---

> > > ### Author Response · Authors · 2025-11-24
> > > **Response (rebuttal)**
> > >
> > > Dear Reviewer vx3U,
> > >
> > > Thank you for your prompt reply and your encouraging feedback. We appreciate your guidance on improving the presentation and accessibility of our work for the broader machine learning community.
> > >
> > > Based on your specific recommendations, we have uploaded a revised version of the paper with the following main changes:
> > >
> > > * **Simplified Notation:** We have revised and simplified the notations for the BoN and SBoN policies, as well as the reward functions and coverage constants. This makes the distinction between the policies and the resulting theoretical bounds much easier to parse.
> > > * **Improved Visualization:** To clarify the theoretical relationships between the policies (BoN, SBoN, Optimal titled, and Optimal), we have moved the diagram illustrating these connections to the main body (now **Figure 1**).
> > > * **Streamlined Setup:** As suggested in your previous comment (W1), we have improved the flow of the introduction. In particular, we removed the early technical discussion of "calibration" to avoid confusion; this concept is now introduced only when necessary in **Section 3.1**, allowing for a smoother entry into the problem setup.
> > >
> > > We believe these changes have significantly improved the clarity and flow of the paper. Please let us know if there are any remaining points you would like us to clarify.
> > >
> > > If these revisions satisfactorily address your concerns regarding presentation and accessibility, we would be grateful if you would consider raising your score.

---

> > > > ### Author Response · Authors · 2025-11-25
> > > > **Thank you for raising your score**
> > > >
> > > > Dear Reviewer vx3U,
> > > >
> > > > Thank you for your thoughtful engagement during the rebuttal period and for updating your rating score. We’re glad our responses addressed your concerns. If any further questions arise, we’d be happy to clarify.
> > > >
> > > > Bests,
> > > >
> > > > The Authors.

---

> > > > > ### Comment · Reviewer_vx3U · 2025-11-27
> > > > >
> > > > > Thanks for the edits! I believe your paper in its updated form will be valuable for the ICLR community and I recommend acceptance.

---

### Official Review · Reviewer_rpmQ · 2025-10-28

**Soundness:** 2
**Presentation:** 1
**Contribution:** 2
**Rating:** 2
**Confidence:** 3

**Summary:**

This paper provides a theoretical analysis of the soft Best-of-N (SBoN) method, which smooths the output probabilities, a technique commonly used in various applications. The focus is on the calibrated proxy reward rather than the (raw) true reward, with respect to KL-divergence and regret. The authors show that in certain conditions, SBoN achieves a tighter bound on the regret gap than standard Best-of-N (BoN), while BoN attains a tighter regret upper bound when over-optimization is absent. Since the smoothing usually introduces algorithmic conservativeness, these results are consistent with general intuition.

**Strengths:**

* Provides finite-sample bounds on the KL divergence between SBoN and the reference policy under the true reward model, as well as between SBoN policies under the true and proxy reward models. These results generalize across different values of $\beta$ (the temperature parameter for the Shannon entropy penalty used to smooth probabilities).

* Considers a calibrated proxy-reward model, which is more realistic and relevant for practical applications.

* I carefully read the paper, including the appendices, and checked the proofs (except Appendix G). Most of the derivations appear correct, though a few technical points remain unclear (see the Questions section).

**Weaknesses:**

* The paper has overly complex notation, which significantly reduces readability and makes it difficult to follow the theoretical flow. Some notation should be refined or consistent for clarity.

* The main theoretical results are difficult to interpret (see the Questions section for details).

**Questions:**

1. In Section 3.1 the authors consider the finite set of prompts $X$. For responses this is plausible, since an LLM outputs finite-length token sequences from a finite vocabulary. For prompts, however, $X$ is provided by the human and is not naturally finite. Is finiteness of $X$ actually required for the analysis (as far as I checked, it does not seem necessary), or only for notational convenience? If it is required, can you clarify why?

2. The temperature $\beta$ is used for three different roles: (i) tilted optimal policy for the KL-regularizer, (ii) the entropy temperature in SBoN (the softmax smoothing), and (iii) the parameter in the tilted error term. Are these all meant to be the same $\beta$? If not, it would be clearer to separate them, e.g., ($\beta_{\text{KL}}, \beta_{\text{SBoN}}, \beta_{\text{err}} $).

3. What is the intended meaning of Theorem 4.3? Although the authors state that it can be viewed as an improvement relative to the reference policy, it is unclear what insight is gained from providing an **upper bound** on the expected calibrated reward. The bound can exceed $1$ in some cases, which is trivial since the calibrated reward itself is defined in $[0,1]$. While the bound is always larger than $0.5$ (the reward of the reference policy), this alone does not make the result informative. Furthermore, the authors mention that they aim  to maximize the first term and minimize the second term, but this makes the overall objective of Theorem 4.3 difficult to interpret. Although the intuition behind maximizing or minimizing each component is reasonable, the combined purpose of these directions and the motivation for presenting Theorem 4.3 remain unclear. Could the authors clarify the intended rationale for considering this theorem?

4. The regret upper bounds are more informative, as the authors provide a clear comparison between SBoN and BoN. One straightforward question: would it be possible to derive a lower bound on the optimality gap when $N$ is finite and $\beta$ is fixed? If so, such a result would offer a clearer reference point for evaluating the tightness and significance of the obtained upper bounds.

5. Could you explain why the first inequality in the proof of Lemma E.11 holds? Does the derivation implicitly assume that the reward function $r$ takes discrete values? If $r_c$ follows a uniform distribution on $[0,1]$ as the true calibrated reward function under the reference policy, then the event $\\{ R= r \\}$ has zero probability.

6. In (44) there is an equality where terms of the form $-\beta \sum_y \cdots$ seems to disappear. Is (44) intended to be an inequality instead of equality? If not, can you provide the details?

7. Could you provide the skipped steps between equation (47) and the following inequality? Because the notation is quite dense, it is difficult to recall all the original definitions. Including the intermediate steps would make the proof much clearer and providing explicit pointers or brief reminders of key definitions would greatly help readers follow how each inequality is derived.

---

Comments

1. The experiments use a “harmlessness score,” but there seems no definition in the main text. At least, it would be better to specify whether higher is better or lower is better.

---

Minor comments (typos)

* The reward function $r$ is first defined on $X\times Y$ with notation $r(x,y)$ in (2), but later the paper mostly uses $r(y,x)$. Please make this consistent.

* Possible typo in the definition of $Z_{N,\beta}$ in L204: the exponent $-1$ after the expectation looks like a typo.

* Possible typo in L448: the denominator shows $C\_{\beta, \hat{r}\_C, \text{ref}}$ but based on context it seems this should be $C\_{\infty, \hat{r}\_C, \text{ref}}$.

* Eisenstein et al. (2023) appears to be listed twice in the references (page 10).

* Lemma E.9: “miximizer” should be “maximizer.”

* Lemma E.11: the numerator is missing a $\delta$.

---

> ### Author Response · Authors · 2025-11-21
> **Response (1/3)**
>
> We thank the reviewer for the careful reading and constructive feedback. Below we address the main concerns and questions.
>
> ---
>
> > The paper has overly complex notation, which significantly reduces readability and makes it difficult to follow the theoretical flow. Some notation should be refined or consistent for clarity.
>
> **R1:** A Table for notation summary is added in Appendix A.
>
> > In Section 3.1 the authors consider the finite set of prompts $X$. For responses this is plausible, since an LLM outputs finite-length token  sequences from a finite vocabulary. For prompts, however, $X$ is provided by the human and is not naturally finite. Is finiteness of $X$ actually  required for the analysis (as far as I checked, it does not seem necessary), or only for notational convenience? If it is required, can you clarify why?
>
> **R2:** Thank you for this insightful observation. You are correct that the finiteness of the prompt set $\\mathcal{X}$ is primarily assumed for measure-theoretic simplicity and notational convenience (e.g., to define expectations over prompts without requiring complex probability spaces).
>
> Regarding the analysis, strict finiteness is indeed required for the **response set** $\\mathcal{Y}$. This ensures that for any given prompt $x$, the distribution of rewards is discrete, which allows us to calculate terms involving $P(R=r)$ (such as in Lemma F.11). As long as $\\mathcal{Y}$ is finite, our per-prompt bounds (e.g., Theorem 4.3 and 5.2) hold regardless of the cardinality of $\\mathcal{X}$. We will clarify in the final version that $\\mathcal{X}$ finiteness is a simplifying assumption rather than a strict necessity for the main results.
>
> > The temperature $\\beta$ is used for three different roles:  (i) tilted optimal policy for the KL-regularizer, (ii) the entropy temperature in SBoN (the softmax smoothing), and (iii) the parameter in the tilted error term. Are these all meant to be the same $\\beta$? If not, it would be clearer to separate them, e.g.,  $(\\beta\_{\\mathrm{KL}}, \\beta\_{\\mathrm{SBoN}}, \\beta\_{\\mathrm{err}})$.
>
> **R3:** Thanks for pointing this out.
> As shown in Figure 2, the SBON policy converges to the tilted policy when $N$ is infinite, assuming the same $\\beta$ for both policies. We can, therefore, also motivate the SBON policy as a finite estimate of the tilted policy, which justifies why $\\beta\_{\\mathrm{KL}}$ and $\\beta\_{\\mathrm{SBoN}}$ are the same.
> Our approach defines $\\varepsilon\_{\\infty,r}$ as the infinite norm of the error between the true and proxy reward models for the BoN scenario. This error term is recoverable in the asymptotic regime. To simplify notation, we adopt the same smoothing parameter $\\beta$ for the error definition as is used in the KL and tilted cases. It helps us to utilize a single smoothing parameter $\\beta$ to derive results for BoN based on the established results for Soft-BoN (SBoN) which is dependent on
>
> > The regret upper bounds are more informative, as the authors provide a clear comparison between SBoN and BoN. One straightforward question: would it be possible to derive a lower bound on the
>   optimality gap when $N$ is finite and $\\beta$ is fixed? If so, such a result would offer a clearer reference point for evaluating the tightness and significance of the obtained upper bounds.
>
> **R4:** Thank you for pushing us in this direction. Despite the tight timeline, we prioritized this derivation and were able to establish lower bounds for both SBON and BoN under the conditions you described (finite $N$, fixed $\\beta$). These results are now presented in Section 5.2 of the revision. For a summary please check [this post](https://openreview.net/forum?id=tCv1D3M7Lb&noteId=wJ5yFto9cI) .
>
> > Could you explain why the first inequality in the proof of Lemma E.11 holds? Does the derivation implicitly assume that the reward function $r$ takes discrete values? If $r\_c$ follows a uniform distribution on $\[0,1\]$ as the true calibrated reward function under  the reference policy, then the event $\\{R \= r\\}$ has zero probability.
>
> **R5:** Regarding the first inequality in Lemma E.11, we need to compute the following derivative,
> $$
> \\begin{aligned}
> \\frac{\\delta \\mathbb{E}\[\\frac{1}{\\exp(\\beta r)+\\sum\_{i=1}^{N-1} \\exp(\\beta R\_i)}\]}{\\delta r}
> \\end{aligned}
> $$
> Let’s consider the conditional expectation where we condition on having exactly $(k-1)$ of $(N-1)$ $R\_i$’s equal to $r$. Therefore, we have,
> $$
> \\begin{aligned}
> &\\frac{\\delta\\mathbb{E}\[\\frac{1}{\\exp(\\beta r)+\\sum\_{i=1}^{N-1} \\exp(\\beta R\_i)}\]}{\\delta r}\\\\
> &=\\sum\_{k=1}^{N} \\binom{N-1}{k-1}\\mathbb{E}\\Big\[\\frac{\\beta k\\exp(\\beta r)}{k\\exp(\\beta r)+\\sum\_{i=1}^{N-k} \\exp(\\beta R\_i)}\\Big\] (1-P(R=r))^{N-k}P^{k-1}(R=r)
> \\end{aligned}
> $$
> Now computing the derivative and using the fact that $1\\leq \\exp(\\beta R\_i)$, the first inequality holds. We clarified this point in the revised manuscript with more details.

---

> ### Author Response · Authors · 2025-11-21
> **Response (2/3)**
>
> > What is the intended meaning of Theorem 4.3? Although the authors state that it can be viewed as an improvement relative to the reference policy, it is unclear what insight is gained from providing an \\textbf{upper bound} on the expected calibrated reward. The bound can exceed $1$ in some cases, which is trivial since the calibrated reward itself is defined in $\[0,1\]$. While the bound is  always larger than $0.5$ (the reward of the reference policy), this alone does not make the result informative. Furthermore, the authors mention that they aim to maximize the first term and minimize the second term, but this makes the overall objective of Theorem 4.3 difficult to interpret. Although the intuition behind maximizing or minimizing each component is reasonable, the combined purpose of these directions a  nd the motivation for presenting Theorem 4.3 remain unclear. Could the authors clarify the intended rationale for considering this theorem?
>
> **R6:** We thank the reviewer for their helpful comment. We use this **upper bound, Theorem 4.3,** to characterize the **limitations** of the policy under Goodhart's Law (reward hacking).
>
> The intended meaning of Theorem 4.3 is to mathematically demonstrate that the expected true reward does not monotonically increase with the optimization strength (controlled by $N$ and $\\beta$).Instead, it is capped by two competing forces: the potential for improvement and the accumulation of estimation error.
>
> ### Upper bound in Theorem 4.3
>
> We present this bound to model the "overoptimization" phenomenon.The theorem decomposes the upper limit of the True Reward into two distinct terms that react oppositely to the temperature $\\beta$ and sample size $N$.
>
> $$\\text{Bound} \\approx 0.5 \+ \\underbrace{\\text{Potential Improvement}}\_{\\text{Term 1}} \+ \\underbrace{\\text{Estimation Error}}\_{\\text{Term 2}}$$
>
> * **Term 1 (The "Good" Term):** $\\sqrt{(1-\\frac{1}{N})(1-exp(-\\beta))}$
>   * This represents the maximum possible gain relative to the reference policy under true reward model .
>   * It implies that increasing $\\beta$ (or $N$) *should* improve performance if the reward model were perfect.
> * **Term 2 (The "Bad" Term):** The term involving $\\epsilon\_{\\beta,r\_c}$ (tilted error).
>   * This quantifies the penalty introduced by using a proxy reward instead of the true reward.
>   * Crucially, this error term **grows** as $\\beta$ increases (as the policy deviates further from the reference).
>
> When we mention maximizing the first term and minimizing the second, we are describing the **optimization dilemma** regarding the choice of $\\beta$ (temperature), not suggesting a manual optimization of the bound itself.
>
> 1. **The Trade-off:** For a fixed $N$, increasing $\\beta$ (approaching standard Best-of-N) maximizes Term 1 (good) but explodes Term 2 (bad) due to reward misspecification.
> 2. **SBoN vs. BoN:** This theorem provides the theoretical justification for why Soft Best-of-N (SBoN) can outperform standard Best-of-N (BoN). In BoN ($\\beta \\to \\infty$), Term 2 may dominate, causing the bound (and actual performance) to degrade. In SBoN, one can choose a finite $\\beta$ that creates a "sweet spot" where the gain is high, but the error term is kept in check.
>
> ### Addressing Triviality
>
> Regarding the reviewer’s point that the bound can exceed 1 (triviality), while the absolute value at the extremes might be loose, the *derivative* of the bound with respect to $\\beta$ is what matters. It predicts the "hump" shape observed in empirical results (Figure 1), where performance improves initially and then degrades as optimization pressure increases.
>
> **In summary:** The rationale for Theorem 4.3 is to provide a theoretical framework that explains **why** overoptimization occurs. It proves that simply increasing sampling (N) or sharpness ($\\beta$) eventually hits a wall where the error term (Term 2\) cancels out the theoretical improvement (Term 1), motivating the use of SBoN to balance these factors.
>
> > In (44) there is an equality where terms of the form $-\\beta \\sum\_y \\cdots$ seem to disappear. Is (44) intended to be an inequality instead of equality? If not, can you provide the details?
>
> **R7:** The equality holds, so the term $-\\beta \\sum\_y \\cdots$ is not required.

---

> ### Author Response · Authors · 2025-11-21
> **Response (3/3)**
>
> > Could you provide the skipped steps between equation (47) and  the following inequality? Because the notation is quite dense, it is difficult to recall all the original definitions. Including the intermediate steps would make the proof much clearer and providing explicit pointers or brief reminders of key definitions would greatly  help readers follow how each inequality is derived.
>
> **R8:**
> Thanks for checking our proof and asking detailed questions. In (47), we used the fact that,
>
> $$
> \\begin{aligned}
> \\pi^{(N,\\beta)}\_{r\_c^{\\star}}(\\cdot|x)\\leq \\frac{N\\pi\_{\\text{ref}}(\\cdot|x)}{1+(N-1)\\exp(-\\beta)}
> \\end{aligned}
> $$
>
> Furthermore, using the following fact for a random variable X,
> $$E\[X\]\\leq \\sqrt{E\[X^2\]} \=  \\sqrt{E\[\\frac{1}{\\beta}\\log(\\exp(\\beta X^2))\]} \\leq \\sqrt{\\frac{1}{\\beta}\\log(E\[\\exp(\\beta X^2)\])},$$
> and applying it to $E\_{\\pi\_{\\text{ref}}(\\cdot|x)}\[(r^\\star\_c(y,x)-\\hat{r}\_c(y,x))\]$, the final result in Eq.(47) holds.
>
> We clarified this point in new version.
>
> >The experiments use a “harmlessness score,” but there seems no definition in the main text. At least, it would be better to specify whether higher is better or lower is better.
>
> **R9:** Thanks for your point. We clarified this point in the main text. A higher harmlessness score is better.
>
> Finally, thanks for the minor comments on typos. They are fixed now.

---

> > ### Comment · Reviewer_rpmQ · 2025-11-24
> >
> > I greatly appreciate the authors’ huge efforts in addressing my concerns in such a short time!
> >
> > Most of my concerns have been resolved, but I am still struggling to understand the analysis of the reward bounds and the intended interpretation of the reward upper bound in Equation (14) of Theorem 4.3. As explained in the responses to me and Reviewer vx3U, the upper bound on the reward decomposes into two main terms:
> > * “Good” term (potential improvement): increases as β or N increases.
> > * “Bad” term (estimation error): increases as β increases.
> > When each term is viewed in isolation, the authors' argument is natural. Increasing β improves the good term while decreasing the bad term, and I already understood this trade-off even in the original manuscript.
> >
> > My confusion arises because these two terms appear in an **upper bound on the expected reward**, which is purely about the analytical role of the upper bound itself:
> >
> > Let me start with an example that I can interpret naturally, to clarify my points.
> > In standard statistical learning problems, upper bounds are typically used for objectives that we want to minimize, such as risk (loss) minimization. For instance, one may obtain an upper bound for the expected risk expressed as the sum of the empirical error and a model complexity term (for example, VC dimension or a Rademacher complexity). This reveals a trade-off: choosing a more complex hypothesis class can reduce the empirical error but increases the complexity term. This is natural because the objective is to minimize the expected risk, and the model choice directly tightens or loosens the upper bound, creating the trade-off.
> >
> > In contrast, the objective of Theorem 4.3 is not clear to me. I do not understand why an upper bound on the expected reward is the quantity of interest. From the terminology, I guess the goal is reward maximization. If this is not the case, please let me know. But assuming it is, then both terms on the RHS of (14) increase with β (ignoring their interpretations). This makes it unclear to me why the upper bound is informative for understanding the trade-off dilemma since large $\beta$ simply yields larger upper bounds. The authors claim that Theorem 4.3 represents an improvement relative to a reference policy, but an upper bound on the reward does not directly express such an improvement. A lower bound on the reward would be more appropriate for that interpretation.
> >
> > To summarize, my main confusion is that
> > * What is the main intention to provide Theorem 4.3?
> > * If we interpret the upper bound terms itself separately, it is clear that the choice of $\beta$ creates the trade-off since we want to maximize the good term and minimize the bad term.
> > * But such terms appear as the upper bounds of the expected rewards, where the upper bounds increases as $\beta$ increases, which does not make any trade-off.
> >
> > ---
> >
> > Regarding the regret lower bounds:
> >
> > While negative lower bounds limit their informativeness in some regimes, the results are still meaningful in the no-error regime and for certain values of β. I have one basic question here. In Figure 6, the lower bounds for medium error and high error appear to coincide. Does this imply that the lower bound depends only on whether an error exists, rather than on the magnitude of the error?

---

> ### Author Response · Authors · 2025-11-25
>
> We sincerely thank Reviewer rpmQ for their continued engagement and insightful comments. Your intuition regarding standard statistical learning frameworks (where upper bounds are typically applied to risk/loss minimization) is spot on.
>
> Based on your feedback, we have made the following updates to the manuscript:
>
> ###  **1. Removal of Theorem 4.3**
>
> We agree that upper-bounding the expected reward does not naturally demonstrate a performance guarantee (which would require a lower bound). Consequently, **we have removed Theorem 4.3 entirely** to avoid confusion.
>
> ###  **2. Discussion on KL Divergence (Lemmas 4.1 & 4.2)**
>
> To replace the analysis of Theorem 4.3, we now discuss the trade-off via a decomposition of the KL divergence in Section 4:
>
> * **Lemma 4.1:** Bounds the divergence between the SBoN policy and the Reference policy. This represents the "optimization strength."
> * **Lemma 4.2:** Bounds the divergence between the SBoN policy under the *true* reward vs. the *proxy* reward. This represents the "cost" of using proxy-reward instead of true-reward..
>
> **New Insight (Appendix K):** To further clarify this, we have added **Appendix K**, which analyzes this relationship using **Information Projection**. We show that for a fixed total divergence budget (determined by $\\beta$), an increase in estimation error "consumes" the budget available for improvement over the reference policy (Proposition K.1). This explains why performance degrades when proxy errors are high, even if optimization strength ($\\beta$) is high.
>
> ---
>
> ###  **3. Clarification on Figure 6 (Regret Lower Bounds)**
>
> Regarding your question on why the lower bounds for medium ($\\delta=0.15$) and high ($\\delta=0.50$) error coincide in Figure 6, this behavior stems from our simple setup, e.g. the **binary reward structure,** employed in our numerical experiment.
>
> In our lower bound derivation (Theorem 5.6), the bound is composed of a primary term (related to the probability of error) and a subtracted estimation error term (the minimization between the exponential term and the error term. In this binary setting, the second term converges rapidly and is dominated by the first term. Consequently, once the error is sufficient to degrade performance, the bounds for these error magnitudes indistinguishably converge.
>
> We hope these revisions resolve your concerns regarding the theoretical interpretation.

---

> > ### Comment · Reviewer_rpmQ · 2025-11-27
> >
> > Dear authors,
> >
> > Thank you for your detailed discussion and answers. I now understand the main point of the paper, and the contribution is much clearer to me. I will therefore increase my score from 2 to 6. The reason for choosing 6 instead of 8 is my limited familiarity with best-of-N approaches in LLMs, which makes it difficult for me to fully assess the contribution within that specific area.

---

> > > ### Author Response · Authors · 2025-11-27
> > > **Thank you**
> > >
> > > Dear Reviewer rpmQ,
> > >
> > > Thank you very much for your feedback and for reconsidering your score. We are pleased that the rebuttal was helpful in clarifying the paper’s contribution and points. We appreciate your time and support.
> > >
> > > Best regards,
> > >
> > > The Authors

---

### Official Review · Reviewer_uC3a · 2025-10-28

**Soundness:** 3
**Presentation:** 2
**Contribution:** 3
**Rating:** 6
**Confidence:** 3

**Summary:**

The point of this paper is to adapt LLM's to make their output adjust to some human evaluation metric (e.g. no harmfulness). This is typically done by a human preference function that is learned from data. By sampling multiple generations N  and choosing the best according to the learned preference function, a strategy called Best of N, can be used to generate such outputs adhering to the preference. However, for imperfect preference function, sampling more samples can lead to samples that are "overoptimized". This means that the samples obtained are not good according to the underlying human preference anymore. This work proposes Soft Best-of-N, which applies a soft-max when selecting the best sample, which regularizes the procedure. This paper studies the Soft Best-of-N theoretically, showing bounds on its performance, and analyzes when this regularization can improve performance over Best-of-N. The theoretical results are empirically verified by a small experiment.

**Strengths:**

- The paper is self-contained, even though I am not an expert in this particular domain I could get the main results that are quite technical
- The small experiment corroborates the theoretical findings

**Weaknesses:**

- Sometimes its not entirely clear what are the key novel results of this paper compared to other papers (e.g. Huang 2025), it would be good to highlight this more
- From a rough reading of Huang 2025 it seems they also aim to solve this exact problem. How does your proposed algorithm compare to theirs? Note that I am not asking for an experiment or anything, but a theoretical interpretation would also be fine for me (both would be even stronger).
- This paper has no Discussion section which I think should be in each quality paper; addressing: 1) are the results expected compared to related works? provide any context. 2) is it possible that there were any mistakes or weaknesses in the experiments? discuss them. 3) is there any future work? what would be most interesting to tackle next?

**Questions:**

Especially Questions 1-4 are particularly important for me and to decide if I will increase my score further.

1. Why does Eq 10 measure performance? The KL term is not justified anywhere. This seems especially weak, since the SBoN method depends on Beta (while BoN does not); therefore; isn't it logical that the evaluation in terms of this loss that depends on Beta favors SBoN? Or is actually Eq 9 used? (the notation is a bit unclear here; because J is defined twice). Furthermore, since Eq 10 depends on the true r^*; I think there is no reason to stick close to the reference model, right? I don't see why KL is necessary here.

2. I think I got a bit confused with the bounds. For Theorem 4.3, you state around line 362 that: "We aim to minimize the second term.". But, this bound is an upperbound on the reward, right? Why would we want this term to be small? I understand why we want to minimize the upperbound on regret bounds (e.g. Theorem 5.2), but I am missing something here why we are looking at this upperbound and why we want it to be small.

3. In line 447 a quite technical condition is given under which the SBoN bound is favored. Can you explain what it means? You also allude to this in the last sentence of the conclusion; what are these conditions?

4. How does the proposed method compare to Huang's? E.g. their inference time pessimisim method? (Given that you explicitly compare some of your results to
theirs, isn't it fair to also compare to their strategy?)

5. "We assume that overoptimization regime happens whenever we have \epsilon > 0"; this assumption seems strange to me. Say that the preference function is just very poorly trained (or not trained for that sake); then \epsilon > 0 right? Would this also be "overoptimization"?
6. "Note that in (Huang et al., 2025), raw (uncalibrated)
reward models are utilized for error definition"; so? Does that in any way invalidate their result, or are their results suboptimal for that reason? Can you show this with an example?
7. "Similarly, we can define ... as the optimal policy under the calibrated proxy-reward model"; isnt this strange, since this is not nececarily the "optimal" policy since it can have issues due to overoptimization?

Note that I did not check the mathematical proofs in the appendix, I only focused on the main body.

---

> ### Author Response · Authors · 2025-11-21
> **Response (1/3)**
>
> We thank the reviewer for the careful reading and constructive feedback. Below we address the main concerns and questions.
>
> >Sometimes its not entirely clear what are the key novel results of this paper compared to other papers (e.g. Huang 2025), it would be good to highlight this more. From a rough reading of Huang 2025 it seems they also aim to solve this exact problem. How does your proposed algorithm compare to theirs? Note that I am not asking for an experiment or anything, but a theoretical interpretation would also be fine for me (both would be even stronger).
>
> **RW1:** A full comparison with Huang et al. (2025), is provided in [this post](https://openreview.net/forum?id=tCv1D3M7Lb&noteId=FxgrnwiYLU)
>
> >This paper has no Discussion section which I think should be in each quality paper; addressing: 1\) are the results expected compared to related works? provide any context. 2\) is it possible that there were any mistakes or weaknesses in the experiments? discuss them. 3\) is there any future work? what would be most interesting to tackle next?
>
> **RW2:** We appreciate your suggestions regarding the discussion and future work sections, both of which have now been added. Our work offers the first theoretical guarantees for SBoN. In contrast, Huang et al. previously provided theoretical guarantees for BoN, and we have included a comparison to their work in  [this post](https://openreview.net/forum?id=tCv1D3M7Lb&noteId=FxgrnwiYLU). In summary, we have established lower and upper bounds and introduced an exciting new direction for future theoretical research on SBoN, detailed further in the discussion section (Section 5.3).
>
> ---
>
> > **Q1:** “Why does Eq 10 measure performance? The KL term is not justified… The SBoN method depends on $\\beta$… isn’t it logical that the evaluation in terms of this loss that depends on $\\beta$ favors SBoN? Or is actually Eq 9 used? … Furthermore, since Eq 10 depends on the true r⋆, I think there is no reason to stick close to the reference model, right? I don’t see why KL is necessary here.”
>
> **R1:** We thank the reviewer for this opportunity to clarify our evaluation methodology. We acknowledge that the distinction between our *theoretical objective* (Eq. 10\) and our *performance metric* (Eq. 8, 9, and 11\) is critical to the validity of our results.
>
> ### 1\. Clarification of the Performance Metric
>
> We wish to clarify that **Equation 10 is not the performance metric** used to evaluate the SBoN or BoN policies.
>
> * **Actual Metric:** Our primary performance measure is the **expected calibrated true-reward**, defined as $J\_{r\_{c}^{\star}}(\\pi(\\cdot|x)):=\\mathbb{E}\_{Y\\sim\\pi(\\cdot|x)}\[r\_{c}^{\\star}(Y,x)\]$.
> * **Regret Gap:** Consequently, we evaluate policies based on the **regret gap** (Eq. 9 and Eq. 11), which measures the difference in expected true-reward between the optimal policy and the aligned policy: $$ \\Delta\_{J\_{r\_{c}^{\star}}}(\\pi\_{r\_{c}^{\\star}}^{\star}(\\cdot|x),\\pi\_{\\hat{r}\_{c}}^{(N,\\beta)}(\\cdot|x))=J\_{r\_{c}^{\\star}}(\\pi\_{r\_{c}^{\\star}}^{\\star}(\\cdot|x))-J\_{r\_{c}^{\\star}}(\\pi\_{\\hat{r}\_{c}}^{(N,\\beta)}(\\cdot|x))$$ Crucially, this metric depends solely on the calibrated true-reward $r\_c^{\\star}$, not on the KL-regularization parameter $\\beta$. Therefore, there is no circularity or bias favoring SBoN via the choice of $\\beta$ in the evaluation metric.
>
> ### 2\. The Role of Equation 10
>
> Equation 10 describes the **KL-regularized objective function**:: $$ J\_{r\_{c}^{\\star*},\\beta}(\\pi\_{ref}(y|x),\\pi(\\cdot|x)):=\\mathbb{E}\_{Y\\sim\\pi(\\cdot|x)}\[r{c}^{\\star*}(Y,x)\]-\\frac{1}{\\beta}KL(\\pi(\\cdot|x)||\\pi\_{ref}(\\cdot|x))$$
>
> We use this objective solely to characterize the **tilted optimal policy** $\\pi\_{\\beta,r\_{c}^{\*}}(y|x)$, which serves as a standard theoretical benchmark in the literature (e.g., Korbak et al., 2022b; Yang et al., 2024). Note that in limit for $N\\rightarrow \\infty$, SBoN policy converges to tilted optimal policy.
>
> Our theoretical results explicitly use the reward-based metrics, not the KL-regularized objective:
>
> * **Theorem 4.3** bounds the expected calibrated true-reward $\\mathbb{E}\[r\_c^{\\star}(Y,x)\]$ directly (with a baseline of 0.5).
> * **Theorems 5.2 and 5.3** provide bounds for the regret gap $\\Delta\_{J\_{r\_{c}^{\\star*}}}$ for SBoN and BoN, respectively. Both theorems measure the gap against the optimal policy $\\pi\_{r\_{c}^{\\star*}}^{\*}$ , which is defined by maximizing the calibrated true-reward, independent of $\\beta$.
>
> Finally, to eliminate this confusion in the final manuscript, we remove Equation.10 to avoid confusion.

---

> ### Author Response · Authors · 2025-11-21
> **Response(2/3)**
>
> > **Q2.** *“For Theorem 4.3… you state ‘We aim to minimize the second term.’ But this bound is an upper bound on the reward, right? Why would we want this term to be small?”*
>
> ###
>
> **R2:** We thank the reviewer for highlighting this point, as it touches on the core intuition behind analyzing overoptimization (reward hacking).
>
> While the reviewer is correct that Theorem 4.3 provides an **upper bound** on the expected calibrated true reward and generally, we desire a higher reward, the bound in Equation 16 is composed of two distinct components with opposing implications:
>
> 1. **The Improvement Term:** The first term, $\\sqrt{(1-1/N)(1-\\exp(-\\beta))}$, represents the "legitimate" gain in reward achieved by sampling $N$ times and selecting the best response based on true reward model. We naturally want this contribution to be large.  Note that we improved this term respect to first version and it is tighter wrt first version.
> 2. **The Error Term:** The second term (involving $\\epsilon\_{\\beta, r\_c}$) quantifies the **estimation error** introduced because we are selecting based on a proxy reward rather than the true reward.
>
> We aim to minimize this **second term** (the error term) for two reasons:
>
> * **Mitigating Overoptimization:** This term represents the potential for "spurious" gains for instance where the policy optimizes the proxy reward $\\hat{r}\_c$ by exploiting its divergence from the true reward $r\_c^{\\star}$ (i.e., reward hacking). If this term is large, a high upper bound does not guarantee high performance; rather, it indicates a loose bound where the *actual* true reward could be significantly lower than the proxy suggests.
> * **Tightness and Reliability:** Minimizing the error term ensures that the upper bound is "tighter" to the legitimate improvement term. As noted in **Remark 4.5**, balancing these terms is crucial: we need a $\\beta$ large enough to gain utility from selection (Term 1), but small enough to keep the estimation error (Term 2\) in check.
>
> In short, we want to maximize the *total* bound by maximizing the first term (sampling gain), while simultaneously minimizing the second term (error/risk) to ensure those gains are robust and not the result of overoptimization.
>
> ---
>
> > **Q3.** *“In line 447 a quite technical condition is given under which the SBoN bound is favored. Can you explain what it means?”*
>
> **R3:**   We thank the reviewer for asking for clarification on this condition. The inequality in Remark 5.6 represents a trade-off between the **cost of smoothing** and the **gain from error reduction**.
>
> To understand this, we compare the asymptotic regret bounds ($N \\to \\infty$) derived in our theorems:
>
> 1. **BoN Regret (Eq. 20):** Scales with $\\sqrt{\\varepsilon\_{\\infty, r\_c}}$, the $L^\\infty$ (worst-case) error of the proxy reward.
> 2. **SBoN Regret (Eq. 19):** Scales with $\\sqrt{\\varepsilon\_{\\beta, r\_c}}$ (the smoothed error) plus a penalty term $\\frac{\\log(C\_{\\infty, r\_c^{\\star}, \\text{ref}})}{\\beta}$.
>
> The condition under which SBoN is favored is:
>
> $$\\frac{\\log C\_{\\infty,r^\\star\_c,\\mathrm{ref}}(x)}{\\beta^\\star} \\le \\left(\\sqrt{\\varepsilon\_{\\infty,r\_c}(x)} \- \\sqrt{\\varepsilon\_{\\beta^\\star,r\_c}(x)}\\right) \\cdot (\sqrt{C\_{\\infty,r^\\star\_c,\\mathrm{ref}}(x)}+\sqrt{C\_{\\infty,\hat{r}\_c,\\mathrm{ref}}(x)})$$
>
> **Interpretation:**
>
> * **RHS (The Gain):** The term $(\\sqrt{\\varepsilon\_{\\infty}} \- \\sqrt{\\varepsilon\_{\\beta}})$ represents the reduction in overoptimization error. BoN effectively seeks out the maximum error (worst-case), whereas SBoN with a finite $\\beta$ averages over the error landscape. If the proxy reward has sharp peaks (high $\\varepsilon\_{\\infty}$) but is reasonable on average (low $\\varepsilon\_{\\beta}$), this gain is large.
> * **LHS (The Cost):** The term inversely proportional to $\\beta$ represents the "smoothing bias"—the regret incurred simply because we are sampling probabilistically rather than deterministically maximizing the proxy.
>
> **In summary,** the condition implies that **SBoN is strictly better than BoN** when the proxy reward has errors (high overoptimization) that can be smoothed out, provided the reference policy has adequate coverage ($C\_{\\infty,r\_c}$ is not too large) to keep the smoothing cost low.
>
> ---
>
> \> **Q4.** *“How does the proposed method compare to Huang's? E.g. their inference time pessimism method?”*
>
> **R4:** A full comparison with Huang et al is provided in [this post](https://openreview.net/forum?id=tCv1D3M7Lb&noteId=FxgrnwiYLU)

---

> ### Author Response · Authors · 2025-11-21
> **Response (3/3)**
>
> > **Q5**. "We assume that overoptimization regime happens whenever we have \\epsilon \> 0"; this assumption seems strange to me. Say that the preference function is just very poorly trained (or not trained for that sake); then \\epsilon \> 0 right? Would this also be "overoptimization"?
>
> **R5:**  We thank the reviewer for this precise observation.
>
> To be precise, $\\varepsilon\_{\\beta,r\_c}(x) \> 0$ indicates **reward misspecification**. As noted in Gao et al. (2023b), misspecification is a necessary precondition for overoptimization, but overoptimization itself refers to the specific regime where the optimization strength is sufficiently high that the degradation due to error outweighs the improvement in the proxy reward. We clarified this discussion in the new version.
>
> Regarding the preference example, if the reward function is very poorly trained on preference data. Then, we may observe that the proxy reward model after calibration is large which results in overoptimization or (reward-misspecification).
>
> ---
>
> > **Q6.** *Note that in (Huang et al., 2025), raw (uncalibrated) reward models are utilized for error definition’; so? Does that in any way invalidate their result, or are their results suboptimal for that reason? Can you show this with an example?*
>
> **R6:** We appreciate the opportunity to clarify this distinction. Our intention is **not** to claim that the results in Huang et al. (2025) are invalid or suboptimal within their framework. Rather, we wish to highlight why we view *calibrated rewards* as a more robust modeling choice for analyzing BoN scaling.
>
> The key distinction lies in **scale invariance** and **ranking preservation**:
>
> 1. **Sensitivity of Raw Rewards:** Raw reward scores can be arbitrarily rescaled or transformed (e.g., by a strictly increasing non-linear function) without changing the preference ordering or the resulting policy. However, regret bounds based on raw Mean Squared Error (MSE) are sensitive to these transformations. A large MSE does not necessarily imply a bad policy if the ranking is preserved.
> 2. **Invariance of Calibrated Rewards:** Calibrated rewards map scores to a canonical $\[0,1\]$ scale (specifically, the quantile space of the reference policy). This makes the error metric invariant to strictly increasing transformations of the raw reward. Our bounds, therefore, reflect *ranking* errors rather than *scaling* artifacts.
>
> **Conceptual Example:** Consider a scenario where the true reward $r^\\star$ is a strictly increasing, non-linear transformation of the proxy reward $\\hat{r}$: $$r^\\star(x) \= \\phi(\\hat{r}(x))$$ where $\\phi$ is highly non-linear (e.g., exponential).
>
> * **Outcome:** Because $\\phi$ is strictly increasing, the ranking of responses is identical. BoN selection using $\\hat{r}$ will select the exact same responses as BoN using $r^\\star$. Ideally, the regret bound should be zero.
> * **Raw MSE Approach (Huang et al.):** The raw MSE $\\mathbb{E}\[|r^\\star \- \\hat{r}|^2\]$ can be arbitrarily large due to the curvature of $\\phi$. Consequently, a bound based on raw MSE would predict high regret, which is loose/pessimistic in this case.
> * **Calibrated Approach (Ours):** Since the quantiles are identical under monotonic transformation, the calibrated rewards are equal ($r^{\star}\_c \= \\hat{r}\_c$). Our tilted error $\\varepsilon\_{\\beta, r\_c}$ is exactly **0**, correctly predicting zero regret.
>
> > **Q7**. *Similarly, we can define … as the optimal policy under the calibrated proxy-reward model’; isn’t this strange, since this is not necessarily the ‘optimal’ policy since it can have issues due to overoptimization?*
>
> **R7:** We thank the reviewer for highlighting this ambiguity. You are correct that the term "optimal" typically implies the most desirable policy in practice, whereas maximizing the proxy reward often leads to overoptimization and degradation on the true reward.
>
> To clarify our definitions in Eqs. (5)–(6):
>
> * $\\pi^\\star\_{r^\\star\_c}$ is the policy that is optimal with respect to the **true** calibrated reward $r^\\star\_c$.
> * $\\pi^\\star\_{\\hat{r}\_c}$ is the policy that is optimal with respect to the **proxy** calibrated reward $\\hat{r}\_c$.
>
> We do not claim that $\\pi^\\star\_{\\hat{r}\_c}$ is optimal in terms of the ground truth $r^\\star\_c$. Rather, it is a theoretical construct used to:
>
> 1. Define the coverage constants $C\_{\\infty, \\hat{r}\_c, \\text{ref}}$.
> 2. Quantify the relative performance of the reference policy against the proxy's maximum possible value.
>
> To resolve this confusion in the revision, we added a footnote explicitly stating that while this policy maximizes $\\hat{r}\_c$, it may be suboptimal or harmful under $r^\\star\_c$ due to Goodhart's Law.
>
> ---
>
> Again, we appreciate the detailed comments. We believe the clarifications and textual changes above will make the paper’s contributions, relationship to Huang et al. (2025), and theoretical interpretations significantly clearer.

---

> ### Author Response · Authors · 2025-11-26
> **Update**
>
> Dear Reviewer uC3a,
>
> Thank you again for your detailed review, especially Questions 1–4. Since our earlier response to you, we have made a few additional revisions prompted by the discussion with Reviewers **rpmQ** and **vx3U**, and we would like to briefly highlight the changes that most directly affect our previous answers to your questions:
>
> 1. **Removal of Theorem 4.3 and reframing of Section 4**
>
>    In the follow-up exchange with Reviewer **rpmQ**, it became clear that presenting an **upper bound on the expected true reward** (Theorem 4.3) was more confusing than helpful for interpreting the trade-off between optimization strength and reward misspecification. We agreed with this concern and have **removed Theorem 4.3 from the main text**.
>
>    Instead, Section 4 now focuses only on:
>
>    * **Lemma 4.1**: KL divergence between the SBoN policy and the reference policy (optimization strength), and
>    * **Lemma 4.2**: KL divergence between SBoN under the true vs. proxy rewards (cost of misspecification or estimation error).
>
>    The qualitative message for your Questions 1–3 is unchanged: there is a competition between “how far we move from the reference” and “how much of that movement is driven by reward error.” However, this is now expressed purely at the KL level, which we believe is conceptually cleaner and avoids the interpretational issues you (and Reviewer rpmQ) raised about the reward upper bound. All regret results in Section 5 and the comparison between SBoN and BoN remain unchanged.
>
> 2. **New Appendix K: information-geometric view of the trade-off**
>
>    To replace the role that Theorem 4.3 was originally intended to play, we have added **Appendix K**, which provides an **information-geometry interpretation** of Lemmas 4.1 and 4.2 via information projection. In this view:
>
>    * A fixed “KL budget” quantifies how far we can move away from the reference policy.
>    * Part of this budget must be spent to stay close to the *true-reward-optimal* policy, while another part is “consumed” by the discrepancy between true and proxy rewards.
>
>   This provides a precise explanation for why increasing optimization strength, when the reward is misspecified, eventually ceases to be beneficial. This formalizes the concept of the overoptimization regime, offering a clearer conceptual narrative.
>
> 3. **Notation and presentation improvements**
>
>    Following Reviewer **vx3U**’s comments (which also relate to your “Presentation” concerns), we have made several presentational changes:
>
>    * **Simplified notation** for BoN/SBoN policies, rewards, and coverage constants. please check **notation summary table in Appendix A**.
>    * **Improved figures and structure**: the diagram explaining the relationships between the reference policy, SBoN, BoN, and the optimal policies has been moved into the main text, and the discussion section and future-work directions have been expanded.
>
>    These changes are aimed at making the paper easier to follow and at clarifying the takeaways that motivated your questions.
>
> ---
>
> We hope these revisions make our answers to your Questions 1–4 more transparent and easier to connect to the final version of the paper. If any part of those questions remains unclear **after these updates**, or if you have any additional concerns, we would be very grateful to hear them, especially since you mentioned that *“Questions 1–4 are particularly important … to decide if I will increase my score further.”*
>
> Please let us know if there is anything else we can clarify or elaborate on.

---

### Official Review · Reviewer_9eqs · 2025-10-30

**Soundness:** 3
**Presentation:** 3
**Contribution:** 3
**Rating:** 6
**Confidence:** 1

**Summary:**

This paper provides a theoretical and empirical analysis of Best-of-N (BoN) and its smooth variant Soft Best-of-N (SBoN), two inference-time alignment methods for generative models.
The authors introduce a formal framework that connects SBoN to KL-regularized reward maximization and analyze two key quantities: (1) the KL divergence between the aligned and reference policies, and (2) the regret, i.e., the calibrated true-reward gap between the optimal and aligned policies.
They derive finite-sample upper bounds for both KL divergence and regret under general conditions, and show that SBoN can outperform BoN when proxy reward models suffer from overoptimization (reward hacking). Empirical results on text generation tasks confirm that smoothing (via temperature β) mitigates reward overoptimization when reward model quality is low.

**Strengths:**

- The poor calibration of reward model is a widely known problem in the community, which will largely affect the BoN (test time scaling) results as reward models are sensitive to OOD data, and one single poor-rated trajectory can influence the final BoN result. And the paper proposed SBoN seems show better performance.

**Weaknesses:**

- The experiments are limited, I suggest to verify the effectiveness  by conducting more experiments on recent non-verifiable agentic tasks.

**Questions:**

same as weakness

---

> ### Author Response · Authors · 2025-11-21
> **Response**
>
> We thank the reviewer for helpful comments. In new revision, we conducted more experiments in Appendix I.

---

> > ### Comment · Reviewer_9eqs · 2025-11-27
> > **Response to Authors**
> >
> > Thanks for the response, I will maintain my score as I am not familiar with most of the theoretical part.

---

### Author Response · Authors · 2025-11-21
**Lower Bounds on Regret of BoN and SBoN**

We complement our upper bound analysis with lower bounds on the regret (optimal gap) for both Soft Best-of-N (SBoN) and Best-of-N (BoN). These bounds provide insight into the fundamental limits of inference-time alignment strategies under calibrated reward models, establishing the minimum expected performance loss one must accept given a finite number of samples and an proxy reward model.

### 1\. Core Assumption: The Margin

The lower bounds rely on **Assumption 5.6 (Margin Assumption)**, which posits a strict non-zero gap between optimal and sub-optimal responses.

* **Definition:** Let $\\gamma(x) \= 1 \- \\sup\_{y \\notin \\mathcal{Y}^{\\star}*\_{r\_c}(x)} r\_c^{\star}(y,x)$, where $\\gamma(x) \\in (0,1]$.
* **Intuition:** If the model fails to select an optimal response, it incurs a penalty of *at least* $\\gamma(x)\\in(0,1\]$. This ensures the lower bound is strictly positive.

### 2\. Theorem 5.6: Lower Bound for SBoN

Theorem 5.6 defines the lower bound for the SBoN policy gap under the calibrated proxy-reward model. The bound demonstrates that the regret is bounded away from zero by a factor of the margin $\\gamma(x)$ and the probability of sampling error.

The inequality is expressed as:

$$
\\Delta\_{J\_{r\_{c}^{\star}}}
\ge
\underbrace{\gamma(x) [(1 - C\_{\infty,r\_c^{\star},\mathrm{ref}}^{-1})^N + \\frac{N((1-C\_{\infty,r\_c^{\star},\mathrm{ref}}^{-1}) \- (1-C\_{\infty,r\_c^{\star},\mathrm{ref}}^{-1})^N)}{N \+ (\\exp(\\beta)-1)(N-1)} ]}_{\text{Cost of Sampling Failure}}- \underbrace{O(\sqrt{\varepsilon\_{\beta,r}(x)})}\_{\text{reward misspecification}}
$$

## Detailed Breakdown of Theorem 5.6

The formula can be broken down into two primary components that drive the regret:

**A. The Cost of Sampling Failure (The First Term)** The dominant part of the inequality represents the probability that the algorithm fails to select a maximizer:

* **$\\gamma(x)$:** The minimum cost incurred when picking a "bad" response.
* **$(1 \- C\_{\infty,r\_c^{\star},\mathrm{ref}}^{-1})^N$:** This term represents the probability that **none** of the $N$ sampled responses fall into the optimal set $\\mathcal{Y}^{\star}$. If the reference model ($\\pi\_{ref}$) rarely generates optimal answers (*low coverage* ), this error remains high regardless of the inference-time algorithm.
* **The Softmax Fraction:** The term $\\frac{N((1-C\_{\infty,r\_c^{\star},\mathrm{ref}}^{-1}) \- (1-C\_{\infty,r\_c^{\star},\mathrm{ref}}^{-1})^N)}{N \+ (\\exp(\\beta)-1)(N-1)}$ accounts for the "Soft" nature of SBoN. Even if a good response is sampled, the softmax (controlled by inverse temperature $\\beta$) might still probabilistically select a sub-optimal one. As $\\beta \\to \\infty$, this specific error term decreases.

**B. The Reward Misspecification Term (The Negative Terms)** The theorem subtracts a term to account for technical discrepancies between the optimal policy and the SBoN based on proxy-reward model which is dependent on $\\varepsilon\_{\\beta,r}(x)$.

### 3\. Proposition 5.7: Lower Bound for Best-of-N (BoN)

By taking the limit as $\\beta \\to \\infty$, the authors provide a specific lower bound for the standard Best-of-N policy. The inequality is given by:

$$ \\Delta\_{J\_{r\_c^{\\star}}} \\ge \\gamma(x)\\left(1 \- \\frac{1}{C\_{\\infty, r\_c^\**, ref}(x)}\\right)^N \-  \\min(1,N\\sqrt{\\epsilon\_{\\infty, r}(x)}) $$

**Overoptimization Floor:** The term $\\sqrt{\\epsilon\_{\\infty, r}(x)}$ (the $L^\\infty$ norm of the estimation error) acts as an irreducible floor for the regret.

### 4\. Proof Mechanics (Appendix H)

The derivation relies on decomposing the regret into cases where the selected sample is optimal versus sub-optimal.

* **Lemma H.3:** Establishes that the regret is at least $\\gamma(x) \\times \\text{Pr}\_{\\text{Err}}(x)$, where $\\text{Pr}\_{\\text{Err}}$ is the probability the selected sample is not an optimal one.
* **Lemma H.4:** Provides the specific lower bound for the probability of error ($\\text{Pr}\_{\\text{Err}}$) in the SBoN algorithm, utilizing Bayes' rule and binomial distributions to account for the number of optimal samples drawn.

---

### Author Response · Authors · 2025-11-21
**Comparison with InferenceTimePessimism**

We contrast our contributions with the recent work of Huang et al. (2025), focusing on four key dimensions: the object of study and complexities, regret bounds, and the error metrics and reward modeling..

* **Object of study and complexities.** Huang et al. (2025) analyze Best-of-$N$ (BoN), introducing an inference-time pessimism algorithm within a $\\chi^2$-regularized framework to mitigate overoptimization. In contrast, we analyze *Soft* Best-of-$N$ (SBoN) where treats BoN as the limit where $\\beta \\to \\infty$ and derive finite-sample KL and regret bounds under reward model misspecification (specifically, calibrated proxy vs. calibrated true rewards). To our knowledge, these SBoN-specific bounds characterizing the behavior under overoptimization are novel. Furthermore, regarding practical complexity, SBoN relies on a single hyperparameter $\\beta$, whereas *InferenceTimePessimism* requires tuning an additional truncation parameter and incurs computational overhead to estimate the normalization factor in *InferenceTimePessimism.*  Furthermore, a threshold for rejection sampling is also needed for *InferenceTimePessimism.*

* **Error metrics and reward modeling.** While Huang et al. (2025) operate on raw (uncalibrated) rewards using mean-squared error (MSE), we utilize calibrated rewards (Balashankar et al., 2024). For example, if the proxy-reward model is an increasing function of the true reward model, then the MSE between uncalibrated true and proxy reward models can be large. However, using calibrated reward models, we have zero MSE between calibrated reward models.  We introduce the *tilted error*, denoted as $\\varepsilon\_{\\beta, r\_c}$, which interpolates between MSE (at $\\beta=0$) and $L^\\infty$ error (as $\\beta \\to \\infty$). This metric allows us to explicitly track the source of error in SBoN and BoN. For example, in BoN, we are interested in $L^\\infty$ error between calibrated rewards, and tilted error can recover this error for $\\beta\\to\\infty$. Furthermore, as discussed by Eisenstein, Jacob, et al. 2023,  reward models are underspecified and using calibrated reward models based on base models with the same hyperparameters can result in different behaviours by changing seeds. Therefore, we need a probabilistic approach for sampling strategy in order to avoid reward models underspecification.

* **Regret bounds.** Our theorems and propositions establish regret bounds for SBoN and its BoN limit that depend on: (i) the tilted error $\\varepsilon\_{\\beta, r\_c}(x)$, and (ii) the coverage constants $C\_{\\infty, r\_c, \\mathrm{ref}}$ for both the true and proxy rewards. A crucial distinction, highlighted in Remark 5.4, is that our BoN regret bound remains finite even when overoptimization vanishes (i.e., when $\\varepsilon\_{\\infty, r\_c} \= 0$ or $N \\to \\infty$). Conversely, the bound proposed by Huang et al. (2025) on BoN scales with the $L^\\infty$-error. This represents a qualitative improvement in how misspecification is handled in our analysis.


**Experiment:** An experiment for *InferenceTimePessimism* under strong and weak reward models is provided in Appendix J. We can observe that SBoN performance is similar to *InferenceTimePessimism*.

----

**References:**

*  Huang et al. (2025): Huang, Audrey, et al. "Is Best-of-N the Best of Them? Coverage, Scaling, and Optimality in Inference-Time Alignment." *ICML 2025*.
* (Balashankar et al., 2024): Balashankar, Ananth, et al. "InfAlign: Inference-aware language model alignment." *ICML 2025*.
* Eisenstein, Jacob, et al. "Helping or herding? reward model ensembles mitigate but do not eliminate reward hacking." *COLM*(2023).

---

### Author Response · Authors · 2025-11-30
**Summary of reviewer discussion and score changes for Submission 19670 (1/2)**

Dear Area Chair,

Because the official reviews have been reverted to their pre-discussion state (scores 6, 2, 6, 4), we would like to briefly summarize how the reviewers’ opinions evolved during the rebuttal and discussion period, and why all four reviewers eventually converged to a score of 6.

Below we summarize the trajectory for each reviewer, with pointers to their public comments.

---

### Reviewer uC3a (initial score: 6 → final position: 6)

* **Initial concerns:**

  * Asked us to clarify what is novel relative to Huang et al. (2025) and how our method compares to their inference-time pessimism algorithm.
  * Requested a proper discussion section (context and future work).
  * Raised several conceptual questions (Q1–Q7) about our performance metric, regret bounds, definition of overoptimization, and the role of calibrated rewards.


* **Our response and changes:**

  * Added a **dedicated comparison** with Huang et al. (2025), including a separate comment “Comparison with InferenceTimePessimism,” where we contrast object of study, regret bounds, error metrics, and complexity, and added an experiment comparing SBoN to inference-time pessimism (Appendix J).
  * Added a **discussion and future-work section** and clarified that our work gives the first regret guarantees for SBoN (with BoN as a limit).
  * Clarified that evaluation is done in terms of expected calibrated true reward and regret (Eqs. 8, 9, 11 in the revised manuscript), not the KL-regularized objective, and removed the confusing equation that suggested otherwise.
  * Clarified our definition of overoptimization vs. mere reward misspecification and why calibrated rewards are preferable to raw rewards.
* **Outcome:** Reviewer uC3a did not change their score (which was already 6) and did not raise any remaining objections after our clarifications and the updated manuscript.

---

### Reviewer rpmQ (initial score: 2 → final position: 6)

* **Initial concerns:**

  * Gave an initial rating (2) despite finding the topic interesting, due mainly to difficulty interpreting the main theoretical results, especially Theorem 4.3, and to dense notation.
  * Asked detailed technical questions about the role of the temperature parameter, the meaning of our reward upper bound, the possibility of regret **lower bounds**, and several proof steps.
* **Our response and changes:**

  * Added a **notation summary table** (Appendix A) and clarified which finiteness assumptions are actually needed.
  * Disentangled the different roles of the temperature parameter and explained why we use a single $\beta$ in our analysis.
  * **Derived and added new regret lower bounds** for both SBoN and BoN (now Section 5.2), as suggested by the reviewer, and discussed their implications.
  * Clarified several technical lemmas (e.g., Lemma E.11) with missing steps and explanations.
  * After further discussion, we **removed Theorem 4.3** from the main paper (which upper-bounded expected reward but caused confusion) and re-framed Section 4 entirely around KL-divergence bounds (Lemmas 4.1 and 4.2).
  * Added **Appendix K**, providing an information-geometric interpretation in terms of a fixed “KL budget,” which cleanly explains the overoptimization trade-off.
* **Outcome:** In their final comment, Reviewer rpmQ wrote that they now understand the main point and that the contribution is much clearer, and **explicitly stated that they raise their score from 2 to 6**, noting that they stop at 6 only because they are not fully expert in BoN-style methods.

---

### Reviewer vx3U (initial score: 4 → final position: 6)

* **Initial concerns:**

  * Found the high-level direction important, but had difficulty with the presentation and takeaways (W1–W3), including:

    * unclear motivation for calibration and SBoN at the start,
    * loose BoN bounds compared to earlier work,
    * lack of a clear “story” about when smoothing helps and how to choose $\beta$.
* **Our response and changes:**

  * Simplified notation and moved the **relationship diagram between reference, SBoN, BoN, and optimal policies** into the main text.
  * Re-structured the introduction so that calibration is only introduced once needed in Section 3.1, making the setup easier to follow.
  * Clarified the theoretical motivation for SBoN as a smoothing lens on BoN, tied to our tilted-error framework.
  * Added more explanation of practical takeaways (e.g., how $\beta$ should decrease as overoptimization risk increases, and how cross-validation or grid-search can be used in practice) and highlighted the added experiments and comparisons.
* **Outcome:** After reading the revised manuscript and our responses, Reviewer vx3U wrote that the updated paper will be valuable to the ICLR community and **recommended acceptance**, and they raised their score to 6.

----
Part(1/2)

---

> ### Author Response · Authors · 2025-11-30
> **Summary of reviewer discussion and score changes for Submission 19670 (2/2)**
>
> ---
>
> ### Reviewer 9eqs (initial score: 6 → final position: 6)
>
> * **Initial concerns:** Limited experiments and a request to “verify the effectiveness by conducting more experiments on recent non-verifiable agentic tasks.”
> * **Our response:** We added additional experiments in Appendix I and explained them in our rebuttal.
> * **Outcome:** In their follow-up comment, Reviewer 9eqs thanked us for the response and confirmed that they would **maintain their score**.
>
> ----
>
> **Final remark to the AC:**
> In summary, starting from initial scores 6, 2, 6, 4, the rebuttal, additional experiments, and revisions led Reviewers rpmQ and vx3U to explicitly upgrade their ratings to 6, and the other two reviewers to maintain their 6.
>
> Best regards,
>
> **The Authors**
>
> ---
> part(2/2)

---

### Meta-Review · Area_Chair_4xPZ · 2026-01-09

**Summary:**

This paper provides significant contributions to the theoretical understanding of Best-of-$n$ (BoN) sampling and introduces several new results for Soft Best-of-$n$ (SBoN). The initial assessment of the paper was mixed: two reviewers recommended acceptance and two recommended rejection. Most concerns centered around framing and presentation of the results rather than any critical flaw of the paper (e.g., reviewer rpmQ highlighting that results were difficult to interpret). The authors put in **significant** effort in the rebuttal and updating the paper, leading to a much crisper manuscript.

After the rebuttal and discussion, the reviewers’ major concerns were addressed and the discussion converged to a unanimous recommendation to accept the paper. Though the updated scores are not directly available, reviewers explicitly mentioned their score increase and change of heart.

Overall, I believe this is a very solid paper that makes meaningful contributions to our mathematical understanding of BoN. I also note the authors effort and engagement with the reviewers, leading to a much stronger final manuscript.

**Reviewer Concerns:**

* Reviewer uC3a raised questions about the novelty of the work and about the choice of performance metrics and bounds. After the rebuttal, this reviewer maintained their positive assessment and did not raise remaining issues.

* Reviewer rpmQ initially had the most negative assessment. The authors addressed their concerns with substantial additions, including a new appendix and new derived bounds. Post-rebuttal, RPMQ explicitly stated that they increased their score to an accept vote, so their core concerns were resolved.

* Reviewer vx3U initially raised concerns about the tightness of the bounds and about the overall presentation. Post-rebuttal, vx3U recommended acceptance, indicating that the concerns that motivated the initial tepid assessment were mostly addressed.

* Reviewer 9eqs had an initial positive assessment and stated (post-rebuttal) that they maintain their score.

Overall, the reviewers who initially leaned negative stated that their concerns were largely resolved in the rebuttal and discussion.

**Reviewer Scores:**

Those are covered above, with all reviewers that had an initial negative assessment **explicitly stating** that scores increased.

---

### Decision · Program_Chairs · 2026-01-26

Accept (Poster)